

# The influence of dust optical properties on the colour of simulated MSG-SEVIRI Desert Dust imagery

Jamie R. Banks[1], Kerstin Schepanski[1], Bernd Heinold[1], Anja Hünerbein[1], and Helen E. Brindley[2]

[1]Leibniz Institute for Tropospheric Research, Leipzig, Germany.
[2]Space and Atmospheric Physics Group, and NERC National Centre for Earth Observation, Imperial College London, London, UK.

*Correspondence to:* Jamie R. Banks (banks@tropos.de)

**Abstract.**

Satellite imagery of atmospheric mineral dust is sensitive to the optical properties of the dust, governed by the mineral refractive indices, particle size, and particle shape. In infrared channels the imagery is also sensitive to the dust layer height and to the surface and atmospheric environment. Simulations of mineral dust in infrared 'Desert Dust' imagery from the Spinning Enhanced Visible and InfraRed Imager (SEVIRI) have been performed, using the COSMO-MUSCAT (COSMO: COnsortium for Small-scale MOdelling; MUSCAT: MUltiScale Chemistry Aerosol Transport Model) dust transport model and the Radiative Transfer for TOVS (RTTOV) program, in order to investigate the sensitivity of the imagery to assumed dust properties. This paper introduces the technique and performs initial validation and comparisons with SEVIRI measurements over North Africa for daytime hours during the six months of the Junes and Julys of 2011-2013. Using T-matrix scattering theory and assuming the dust particles to be spherical or spheroidal, wavelength- and size-dependent dust extinction values are calculated for a number of different dust refractive index databases, along with several values of the particle aspect ratio, denoting the particle shape. It is found that spherical particles do not appear to be sufficient to describe fully the resultant colour of the dust in the infrared imagery. Comparisons of SEVIRI and simulation colours indicate that of the dust types tested, the dust refractive index dataset produced by Volz (1973) shows the most similarity in the colour response to dust in the SEVIRI imagery, although the simulations have a smaller range of colour than do the observations. It is also found that the thermal imagery is most sensitive to intermediately sized particles (radii between 0.9 and 2.6 $\mu$m): larger particles are present in too small a concentration in the simulations, as well as with insufficient contrast in extinction between wavelength channels, to have much ability to perturb the resultant colour in the SEVIRI dust imagery.

## 1 Introduction

For over a decade since they were first launched in 2002, the Spinning Enhanced Visible and InfraRed Imager (SEVIRI) instruments onboard the Meteosat Second Generation (MSG) series of satellites (Schmetz et al., 2002) have provided a wealth of high-temporal resolution information on the meteorology, climate and the surface environment of Africa, Europe, the Middle East, and the Atlantic. In geostationary orbit above the Gulf of Guinea, SEVIRI is particularly well-placed to observe the deserts of the Sahara and the Arabian Peninsula, providing detailed information on atmospheric desert dust from and over these desert





regions (e.g., Slingo et al., 2006; Banks et al., 2017). Dust has an impact on the radiation balance of the atmosphere (e.g. Quijano et al., 2000; Allan et al., 2011; Ansell et al., 2014) in both the solar and the terrestrial radiation bands, and as such the perturbations to the radiation balance of the atmosphere due to dust are observable in visible and infrared imaging channels.

Codified by Lensky and Rosenfeld (2008), the 'Desert Dust' RGB (red-green-blue) scheme is designed as a visual aid to highlight the presence of atmospheric desert dust in SEVIRI imagery, using three of the SEVIRI infrared (IR) channels in the window region of the terrestrial spectrum. An example is shown in Figure 1. Commonly also known as the 'Pink Dust' scheme, atmospheric mineral dust in this rendering scheme is characterised by a distinctive pink colour, in contrast with a light blue sandy desert background surface during the daytime. This description of dust in the imagery is however a substantial simplification of the overall picture, which is complicated by the characteristics of the background surface (thermal emissivity and surface skin temperature), the atmospheric state (temperature and water vapour), and the precise characteristics of the dust itself (optical properties and size/vertical distribution). All of these factors will also influence the resultant colour in the imagery (Brindley et al., 2012), not just the quantity of atmospheric dust.

An important application of this imagery is its capability to track desert dust storm outbreaks, given its high temporal resolution. Manual observation of consecutive SEVIRI images allows for the progression of dust storms to be tracked in time and space, from their origins to their eventual dissipation. This is particularly useful for dust source identification (e.g. Schepanski et al., 2007; Ashpole and Washington, 2013), and is an important source of motivation for understanding in detail the capabilities of the imagery technique. For example, can we be certain that the locations where we first observe the pinkness characteristic of a dust storm are actually where the dust storms had their source? The background conditions and the typical dust type affect the image appearance, which means that there are regional differences in the applicability of the imagery: for example, Murray et al. (2016) have recently shown that over southern Africa the Desert Dust scheme appears to be inferior to alternative renderings in terms of identifying precise dust sources and emission times.

During the Fennec campaign in the Junes of 2011 and 2012, a coordinated effort was made to take in situ measurements of the Saharan climate, specifically the Saharan Heat Low (Engelstaedter et al., 2015) and the lofted desert dust within it, from the ground (e.g. Hobby et al., 2013; Marsham et al., 2013; Todd et al., 2013) and from aircraft (e.g. Rosenberg et al., 2014; Ryder et al., 2015; Schepanski et al., 2013). As one of the few measurement campaigns which has reached into the central Sahara, the combined set of Fennec data is unparalleled in its capacity to provide evidence as to the nature of dust activity over the central desert. For example, two ground-based measurement 'super-sites' were specifically established at remote central desert sites at Bordj Badji Mokhtar (BBM) in Algeria and Zouerat in north-western Mauritania. The summer of 2013 brought two more measurement campaigns based in Saharan outflow regions, the ChArMEx (Chemistry-Aerosol Mediterranean Experiment) campaign in the Mediterranean (Mallet et al., 2016), and the SALTRACE (Saharan Aerosol Long-range TRansport and Aerosol-Cloud-interaction Experiment) campaign in the Atlantic between Senegal and Barbados (Weinzierl et al., 2017).

This paper describes a new analysis tool to investigate the SEVIRI Desert Dust imagery, combining output from the aerosol transport model COSMO-MUSCAT (COSMO: COnsortium for Small-scale MOdelling; MUSCAT: MUltiScale Chemistry Aerosol Transport Model) with the Radiative Transfer for TOVS (RTTOV) program to simulate brightness temperatures and hence colours within the imagery. The period considered includes the Junes and Julys of 2011-2013, covering both the Fen-





nec (2011, 2012) and the ChArMEx/SALTRACE (2013) campaign periods. In Section 2 we describe the assumptions and methodology involved in producing the SEVIRI imagery, and the experimental setup combining the COSMO-MUSCAT and RTTOV models. In Section 3 we explore the current knowledge of dust optical properties in the infrared, what their properties are when considered over the SEVIRI infrared channels and COSMO-MUSCAT size bins, and what this may imply a priori

for the imagery. In order to assess the dust aerosol optical depth (AOD) values simulated by COSMO-MUSCAT, in Section 4 we compare these values with ground-based AERONET (Aerosol Robotic Network) measurements and co-located SEVIRI retrievals of AOD. Finally in Section 5 we take a look at the colour output with respect to dust loading and optical properties, and also with reference to measurements and retrievals by SEVIRI. This comparison with the satellite observations provides an assessment of the accuracy and applicability of the modelling approach. In a follow-up paper, we will investigate further the

relationships between dust AOD, optical properties and height, surface and atmospheric properties, and the complex interplay of their effects on the resultant colour.

## 2   Satellite observations and modelling strategy

### 2.1   SEVIRI and 'Desert Dust' RGB imagery

Operated by the European Organisation for the Exploitation of Meteorological Satellites (EUMETSAT) and positioned on

the MSG series of satellites in geostationary orbit above the equatorial east Atlantic ($0°$N, currently at $0°$E), the SEVIRI instruments provide measurements of the Earth's surface and atmosphere every fifteen minutes in three visible/near-infrared channels, eight infrared channels, and one high spatial-resolution broadband visible channel (Schmetz et al., 2002). The eleven standard channels have spatial sampling rates at nadir of ∼3 km. In the IR the channels are centred at ∼ 3.9, 6.2, 7.3, 8.7, 9.7, 10.8, 12.0, and 13.4 $\mu$m: the 6.2 and 7.3 $\mu$m channels are used for measurements of water vapour at different altitudes in the

troposphere, and the 9.7 $\mu$m channel is in an ozone band. The 8.7, 10.8 and 12.0 $\mu$m channels are broadly located within the atmospheric window region, in which the atmospheric transmission of IR radiation is at its maximum, and so these channels are most representative of measurements of the planetary surface or dominant atmospheric feature (ocean, land, cloud, or aerosol).

SEVIRI measurements are made in the form of counts, which are multiplied by instrument and channel-specific gains and offsets to derive the measured radiances $B$ (in $\mathrm{mW\,m^{-2}\,sr^{-1}\,(cm^{-1})^{-1}}$), defined with respect to wavenumber (with units of

cm$^{-1}$). Imagery created using SEVIRI IR data tends to be derived from a re-interpretation of the radiance data in the form of brightness temperatures, the temperature at which the measured channel radiance would be representative of blackbody radiation. The brightness temperature $T_{\mathrm{B}}$ is derived by inverting the Planck function (e.g. Schmetz et al., 2002):

$$T_{\mathrm{B}} = \left( \frac{c_2 \nu}{\ln \frac{c_1 \nu^3}{B} + 1} - b \right) \bigg/ a. \tag{1}$$

The Planck radiation constants $c_1$ and $c_2$ have values $1.19104 \times 10^{-5}\,\mathrm{mW\,m^{-2}\,sr^{-1}\,(cm^{-1})^{-4}}$ and $1.43877\,\mathrm{K\,(cm^{-1})^{-1}}$, while

$\nu$ is the channel central wavenumber (cm$^{-1}$). The coefficients $a$ and $b$ are instrument- and channel-specific, and have been calculated to account for the width of the SEVIRI channel filter functions.



The SEVIRI Desert Dust RGB imagery is defined (Lensky and Rosenfeld, 2008) by the following set of IR channel brightness temperature differences, at 8.7, 10.8 and 12.0 $\mu$m, within ranges of values specified by minimum and maximum values ($Min_{\mathrm{RGB}} \rightarrow Max_{\mathrm{RGB}}$):

$$R = \frac{(T_{\mathrm{B}120} - T_{\mathrm{B}108}) - Min_{\mathrm{R}}}{Max_{\mathrm{R}} - Min_{\mathrm{R}}} \qquad (-4 \rightarrow 2\,K) \qquad (2)$$

$$G = \Big(\frac{(T_{\mathrm{B}108} - T_{\mathrm{B}087}) - Min_{\mathrm{G}}}{Max_{\mathrm{G}} - Min_{\mathrm{G}}}\Big)^{1/2.5} \qquad (0 \rightarrow 15\,K) \qquad (3)$$

$$B = \frac{T_{\mathrm{B}108} - Min_{\mathrm{B}}}{Max_{\mathrm{B}} - Min_{\mathrm{B}}} \qquad (261 \rightarrow 289\,K). \qquad (4)$$

$RGB$ values are bounded within the range of 0 to 1. As with all rendering schemes, different surface types and atmospheric conditions have distinct colour characteristics. A summary of the colour spectrum available for the cloud-free summertime desert environment is provided by Figure 2, which plots the resultant colour as a function of red and green values, assuming that the blue beam is at its maximum value of 1. In the absence of cloud, the blue beam is often maximised over hot desert surfaces, as well as over many other ground surface types present in northern Africa, whereas blue values of 0 are much more indicative of cloud. The difference between the 12.0 and 10.8 $\mu$m brightness temperatures is particularly useful as a measure of the opaqueness of clouds (Inoue, 1987), typically quite negative for thinner clouds but approaching parity for thicker clouds, so for example deep convective clouds are marked by moderate values of $T_{\mathrm{B}120} - T_{\mathrm{B}108}$ (with cold temperatures at 10.8 $\mu$m) and hence are distinguished in the imagery by their dark red colours. Dust has a stronger positive contrast between these two channels than do water or ice clouds (Prata, 1989), so the red beam tends to be enhanced when dust is present.

For sandy desert surfaces there is generally a high positive contrast between the 10.8 and 8.7 $\mu$m brightness temperatures, a consequence of the particularly low emissivity of large sand particles at 8.7 $\mu$m (Wald et al., 1998), leading to high green values over sandy deserts in the imagery. Surface emissivities for the 8.7 $\mu$m SEVIRI channel used in the RTTOV simulations (Section 2.3) are mapped in Figure 3 (Seemann et al., 2008; Borbas and Ruston, 2010) for June, derived from Moderate Resolution Imaging Spectroradiometer (MODIS) satellite data. Meanwhile the surface emissivities at 10.8 and 12.0 $\mu$m show much less spatial variation with minimum values of ~0.93 at 10.8 $\mu$m and 0.94 at 12.0 $\mu$m: the emissivity at 12.0 $\mu$m is very slightly higher, leading to a tendency for the $T_{\mathrm{B}120} - T_{\mathrm{B}108}$ difference to be very slightly positive over desert surfaces and hence moderate red values are present. The combination of moderately high red and green values, and maximum blue values, over desert surfaces leads to characteristic light blue colours for desert sands during sunlit hours. At night and in cooler months, the blue beam can be below its maximum temperature of 289 K, and so pinker or redder colours are more apparent.

By contrast with sandy surfaces, rockier desert surfaces have higher emissivity values at 8.7 $\mu$m, leading to smaller green values, which when combined with high blue and moderate red values leads to more characteristic purple colours. Similarly, smaller lofted dust particles have higher emissivity values than does the sandy desert (e.g. Ackerman, 1997; Wald et al., 1998), hence airborne dust tends to contribute to a suppression of the green beam in the imagery. Considering the overall effect of dust on the beams, it is therefore to be expected that dust boosts the red beam, reduces the green, and leaves the blue beam maximised, a combination which gives rise to distinctive pink colours. Caution is required when discriminating between lofted dust and background rocky desert surfaces, which can have similar characteristics in the reduction of their green beams, leading



to a potentially subtle distinction between pink dust and purple surface, dependent on dust type. Given that some types of dust in various regions of North Africa under various conditions can appear more purple than pink, this is a case where analysing a succession of images can be invaluable in discriminating between stationary surface features and transitory atmospheric dust

(e.g. Ashpole and Washington, 2012). In the absence of cloud, the maximum value of 289 K in the blue beam is a particularly cold value for the summertime Sahara during the daytime, even for lofted dust. The deepest pink colours are therefore produced for maximum red and minimum green.

Moist air is also often visible in the IR imagery over desert regions, particularly in southern areas such as the Sahel, where it is manifested by areas with deeper blue colours than the surrounding drier air masses. Previous simulations have shown (e.g.

Brindley and Allan, 2003) that of the three SEVIRI IR channels of interest, the $12.0\,\mu$m channel has stronger water vapour continuum absorption than the 8.7 and $10.8\,\mu$m channels, so for a given moisture content this channel reduces in brightness temperature by more than do the other two channels. This implies a reduction in the red beam with respect to drier atmospheres, and hence over hot desert surfaces this leads to increases in blue/turquoise colours. Considering the inclusion of dust in moist atmospheres this also implies that for a given dust loading the dust will appear less red, i.e. less pink, under moist rather than

dry conditions. Moisture can therefore obfuscate the presence of dust in IR imagery and retrievals using IR channels (e.g. Brindley et al., 2012; Banks et al., 2013).

The retrievals from SEVIRI of dust aerosol optical depth (AOD) over arid and semi-arid surfaces used here are a derived product (Brindley, 2007; Brindley and Russell, 2009; Banks and Brindley, 2013) which quantifies the optical thickness of dust per SEVIRI pixel, taking advantage of SEVIRI's IR channels over bright desert surfaces. The algorithm takes a multi-step

approach to retrieving AOD, beginning with the detection of cloud and dust (Derrien and Le Gléau, 2005; Ipe et al., 2004) for each pixel: retrievals are then carried out if the dust flag identifies a pixel as dusty, regardless of whether it has also been identified as cloudy, and for clear pixels. For each pixel and timeslot during the day, a 28-day rolling window of 'pristine-sky' values of brightness temperatures at $10.8\,\mu$m is calculated, using a similar method to that proposed by Legrand et al. (2001). These pristine-sky brightness temperatures account for variations in skin temperatures and total column water vapour

as simulated by ECMWF ERA-Interim reanalysis data (Dee et al., 2011). The instantaneous effect of dust on the brightness temperature $\Delta T_{\mathrm{B108}}$ is calculated by subtracting the instantaneous brightness temperature from the pristine-sky value. Finally, the AOD at 550 nm is calculated using a simulated relationship between the AOD at this wavelength and $\Delta T_{\mathrm{B108}}/\Delta T_{\mathrm{B134}}$. The $CO_2$ absorption band which gives rise to $T_{\mathrm{B134}}$ is used as a proxy for the temperature of the atmosphere in the vicinity of the dust layer (Brindley and Russell, 2009).

## 2.2 COSMO-MUSCAT simulations of dust generation and transport

The atmospheric dust life-cycle, describing dust emission, transport and deposition is simulated by the atmosphere-dust model system COSMO-MUSCAT that consists of version 5.0 of the non-hydrostatic atmosphere model COSMO (Schättler et al., 2014), which is coupled online to the 3-D chemistry tracer transport model MUSCAT (e.g. Wolke et al., 2012). The dust life-cycle is described by different modules implemented in MUSCAT (Heinold et al., 2011): dust emission follows the parameterisation developed by Tegen et al. (2002); dust deposition is described following Berge (1997), Jakobson et al. (1997)



and Zhang et al. (2001), distinguishing between dry and wet deposition (in-cloud scavenging and wash-out). Dust transport is described as passive tracer transport. The size dependency of all dust processes is resolved with respect to five dust size bins ranging from 0.1 to 24 $\mu$m in radius (Table 1). Dust emission fluxes are calculated for grid cells (regions) that are identified as active dust sources using the dust source activation frequency (DSAF) dataset (Schepanski et al., 2007). Atmospheric and hydrological conditions simulated by COSMO together with soil characteristics such as soil texture and soil particle size distribution (Tegen et al., 2002), vegetation cover expressed by the leaf area index (LAI), sub-grid scale orography and surface roughness length, determine the actual dust flux for each grid cell. The model setup has already been tested and validated against observations as presented in detail by Schepanski et al. (2016, 2017).

Dust AODs are calculated at 550 nm from COSMO-MUSCAT dust concentrations by assuming the particles to be spherical with the density of quartz ($\rho_\mathrm{p} = 2.65\,\mathrm{g\,cm^{-3}}$):

$$AOD = \sum_j \sum_k \left( \frac{3}{4} \frac{Q_{\mathrm{ext,550}}(j)}{r_{\mathrm{eff}}(j)\rho_\mathrm{p}(j)} c_{\mathrm{dust}}(j,k)\Delta z(k) \right), \tag{5}$$

with dust bin $j$, vertical level $k$, dust particle effective radius $r_{\mathrm{eff}}(j)$, dust concentration $c_{\mathrm{dust}}(j,k)$, and the vertical increment of each level $\Delta z(k)$. The extinction efficiency at 550 nm is calculated from Mie theory (e.g. Mischenko et al., 2002) using refractive indices from Sinyuk et al. (2003) and set to $Q_{\mathrm{ext}} = (1.677, 3.179, 2.356, 2.144, 2.071)$ for the five size bins.

Here, COSMO-MUSCAT simulations are performed for three two-month periods leading to a dataset consisting of six months of data: the Junes and Julys of 2011, 2012 and 2013. To allow for a full two-month dataset for analysis, the simulation is chosen to start mid-May in order to initiate a realistic atmospheric dust distribution for the period of interest. As the focus of this study is on airborne dust over northern Africa, the model domain covers Africa north of the Equator. COSMO-MUSCAT simulations are performed at 28 km horizontal grid spacing and 40 vertical sigma-p levels with the lowest level being centred at 10 m above ground.

## 2.3 RTTOV simulations of SEVIRI brightness temperatures

In order to derive the simulated brightness temperatures from the COSMO-MUSCAT dust distribution simulations, an extra modelling step is required. Here we make use of the EUMETSAT NWP SAF (Numerical Weather Prediction Satellite Application Facilities) radiative transfer model RTTOV (Radiative Transfer for TOVS, Saunders et al. (1999), Matricardi (2005), publicly available at https://nwpsaf.eu/site/software/rttov/). This has previously been used with some success to simulate, for example, volcanic ash in SEVIRI imagery (Millington et al., 2012). As a radiative transfer tool it has been designed to be fast to run, being specifically designed with satellite applications in mind. Within the context of this study, COSMO-MUSCAT provides both the background atmospheric state and the dust concentrations as input into RTTOV, which then calculates the SEVIRI channel radiances and brightness temperatures as would be measured at the top-of-the-atmosphere (TOA). Specific satellite and instrument calibration coefficients are included in these calculations. The background state requires profiles of atmospheric pressure, temperature and humidity, as well as surface type, emissivity (distinct for each SEVIRI channel) and temperature. Solar and satellite viewing zenith angles and azimuthal angles are also required. For the purposes of this study



RTTOV simulations are only performed for land grid cells with COSMO-MUSCAT simulated cloud fractions of less than 1 %, to be unambiguous that it is only dusty or clear scenes that are being simulated.

5 Throughout the rest of this paper, RTTOV simulations performed offline with COSMO-MUSCAT output will be referred to as COSMO-MUSCAT-RTTOV. Simulations of the background atmospheric and surface state which do not include dust are referred to as 'pristine-sky' simulations. Including dust in the simulations requires profiles of the dust absorption and scattering coefficients, in units of $km^{-1}$, as well as profiles of the back-scatter parameter, which defines the integrated fraction of back-scattered energy and which is derived from the overall phase function of the dust layer (e.g. Matricardi, 2005).

## 3 Dust optical properties

Dust optical properties are governed by the refractive indices, the size, and the shape of the dust particles. Refractive indices for a given dust type are wavelength-dependent, and are defined by their real and imaginary parts (e.g., Sokolik et al., 1993; Di Biagio et al., 2014a), $m(\lambda) = n(\lambda) - ik(\lambda)$. The real part ($n$) is the ratio of the speed of light in a vacuum to that in the dust particle (specifically, the phase velocity); meanwhile the imaginary part ($k$, historically also known as the absorption index) is representative of the material's light absorption properties, and hence is a more significant indicator of dust particles' influence on the measured and simulated IR brightness temperatures.

Early work which sought to measure the wavelength-resolved IR refractive indices of dust was carried out by Volz (1973), hereinafter referred to as 'VO73', who analysed samples of transported Saharan dust collected in Barbados. The method applied was the KBr (i.e. potassium bromide) disk technique (Volz, 1972), in which collected dust particles are combined in 20 a dehydrated pellet mixture with potassium bromide, and then spectrographic measurements are performed on these pellets. They noted that the samples of transported dust they were analysing were composed mostly of clay, illite and kaolinite, and also trace amounts of quartz. Significantly, they also pointed out that the dust they took measurements on was particularly absorbing in the atmospheric window region, i.e. at a wavelength of $\sim$8-12 $\mu$m.

Taking a different approach, Sokolik and Toon (1999) collated from previous authors a set of wavelength-dependent IR 25 refractive indices of specific minerals commonly found in arid regions, and proposed a technique to calculate the radiative properties of externally and internally mixed (i.e. aggregates) dust as may be found in the atmosphere above and downwind of desert regions. Minerals of interest included illite, kaolinite, montmorillonite, hematite, quartz, calcite, and gypsum. The core principle of their argument was that it is more robust to measure the optical properties of individual minerals, rather than to measure the optical properties of the mixtures; the optical properties of the minerals can then be composited to retrieve the 30 optical properties for a given specified mixture of minerals.

Helmert et al. (2007) carried out a study on the radiative impact of dusts simulated by COSMO-MUSCAT, both in the short-wave and the long-wave, considering several dust types including the VO73 and Sokolik and Toon (1999) dust types described above. For the dust derived from the Sokolik and Toon (1999) database they considered an internal mixture of 98% kaolinite, 2% hematite. This was derived from a linear interpolation of the Bruggeman approximations carried out by Sokolik and Toon (1999) for mixtures of 90 and 99% kaolinite, with corresponding hematite compositions of 10 and 1%. It is this





database, derived from Sokolik and Toon (1999) by Helmert et al. (2007), that is to be tested in the current study, and which will henceforth be referred to as the 'SO99' dataset.

The 'Optical Properties of Aerosols and Clouds' database, OPAC (Hess et al., 1998), has been widely and diversely used in
atmospheric aerosol studies (e.g., Patadia et al., 2009; Liu et al., 2004; Klüser et al., 2011), with varying degrees of success. It has been used both in modelling and satellite aerosol-retrieval studies, and in both the solar and terrestrial parts of the spectrum. The default dust aerosols simulated by the RTTOV program itself are derived from OPAC (Matricardi, 2005), defined in three lognormal modes of size distribution: the nucleation, accumulation, and coarse modes, with mode radii of 0.07, 0.39, and 1.9 $\mu$m. These modes are not used in the current study however, since the five COSMO-MUSCAT size bins fit very poorly to
these three distributions and any fit that could be attempted would be ill-constrained. Hence only the wavelength-dependent OPAC refractive indices will be considered here, with absorption and scattering coefficients treated in a consistent manner as with the other dust types. It has been noted by, e.g. Di Biagio et al. (2014b), that OPAC dust has a particularly strong imaginary part of the refractive index between $\sim$11-13 $\mu$m compared to other measurements and derivations of mineral optical properties, due to the high quartz content of the assumed dust model. This would have particular implications for the influence of dust on
the red beam of the imagery. Considered in this paper is version 4.0 (Koepke et al., 2015) of the OPAC dataset.

More recent laboratory measurements using samples of suspended dust from various regions of the world have been made by Di Biagio et al. (2017), henceforth denoted DB17, following on from work previously published by the same lead authors (Di Biagio et al., 2014a). In this work (DB17) they sought to measure IR refractive indices from a much wider range of global dust samples, collected in situ around the world but measured in Paris using an experimental measurement chamber,
CESAM (Chambre Expérimentale de Simulation Atmosphérique Multiphasique). Eight of the samples were collected from countries within the Sahara and the Sahel, further samples were collected from arid or semi-arid regions in southern Africa, the Arabian Peninsula, central Asia, Australia, and the Americas. The dried and sieved soil samples were mechanically shaken into suspension in a flask and then injected and dispersed into the chamber, where the dust sample was held in suspension for one to two hours. The dust extinction was measured using a Fourier Transform Infrared (FTIR) spectrometer at a spectral
resolution of 2 cm$^{-1}$, and the size distribution was measured using a scanning mobility particle sizer and two optical particle counters. Combining this information the authors then retrieved the IR refractive indices for each sample using an optical inversion procedure.

Figure 4 summarises the real and imaginary parts of the refractive index for nine of the dust types and mixtures of interest, convolved over the 8.7, 10.8, and 12.0 $\mu$m MSG3-SEVIRI channels. The imaginary part shows a high degree of variability,
both within individual dust types as a function of wavelength, but also between dust types for particular wavelength channels. In contrast there is a greater degree of consistency in the measurements of the real part. The difference between the imaginary part of the refractive index at 10.8 and at 12.0 $\mu$m will influence the intensity of the red beam when dust is present, while the difference between the 8.7 and the 10.8 will influence the intensity of the green. The extinction, however, also depends on the particle size and shape, so it is impossible to state categorically what the influence on the final colour will be from this information alone.





In order to calculate the dust extinction (along with its components, absorption and scattering), we apply T-matrix scattering theory (e.g., Mishchenko and Travis, 1994; Mishchenko et al., 1996) using the database wavelength-dependent dust refractive indices and the effective radii ($r_{\text{eff}}$) of the five COSMO-MUSCAT dust size bins. The particles are here treated as spheres or

spheroids, requiring estimates of typical dust aspect ratios to be input into the T-matrix calculations. The aspect ratio ($AR$) is defined as the ratio of the semi-major to the semi-minor axes of the spheroids. An $AR$ value of 1 indicates spheres. Particle size and wavelength ($\lambda$) are connected through the scattering parameter $x$:

$$x(r_{\text{eff}}, \lambda) = \frac{2\pi r_{\text{eff}}}{\lambda}. \tag{6}$$

T-matrix calculations require the scattering parameter, the complex refractive index, the particle aspect ratio, and the particle

orientation as input, generating extinction, absorption and scattering efficiencies ($Q(r_{\text{eff}}, \lambda)$). Calculations are performed using the Python 'PyTMatrix' code described by Leinonen (2014). The efficiencies for the wavelengths in the dust databases are then convolved over the SEVIRI filter response functions in order to derive the overall efficiencies ($Q(r_{\text{eff}}, ch)$) of dust for the SEVIRI channels, $ch$. Multiplying by the particle cross-sections for the five COSMO-MUSCAT size bins, we derive the extinction, absorption and scattering cross-sections. These time- and space-invariant extinction cross-sections are then multiplied

by the temporally ($t$) and spatially varying ($x$, where $x$ is three-dimensional) particle number densities for each bin to calculate the extinction, absorption and scattering coefficients ($\beta(r_{\text{eff}}, ch, t, x)$), in units of km$^{-1}$ as required by RTTOV. This can also be formulated in terms of the mass concentrations ($M(r_{\text{eff}}, t, x)$, units of kg m$^{-3}$) as output by COSMO-MUSCAT, with respect to the individual particle mass (e.g. Tegen et al., 2010):

$$\beta(r_{\text{eff}}, ch, t, x) = 1000 \times \frac{3\, Q(r_{\text{eff}}, ch)\, M(r_{\text{eff}}, t, x)}{4\, r_{\text{eff}}\, \rho_{\text{p}}}. \tag{7}$$

The AOD is the dust extinction coefficient integrated over the atmospheric dust column. Example mean IR AODs for the central Saharan BBM (Marsham et al., 2013) site in Algeria are plotted in Figure 5, for each of the five size bins and their total, for spherical dust. Bin 5 (radii between 7.9 and 24.0 $\mu$m) contributes very little to the overall extinction due to the typically very small simulated number densities of dust of this size, with AOD values not exceeding $10^{-7}$ for this averaged set of data over BBM. Bin 3 (radii between 0.9 and 2.6 $\mu$m), and to a lesser extent bin 2 (radii between 0.3 and 0.9 $\mu$m), are the major

contributors to the dust signal as expressed in terms of AOD, and hence we would also expect them to have the major influence on the resultant imagery. Bearing this in mind, we note that the greatest contrast in the AODs between the 10.8 and 12.0 $\mu$m channels is in bin 3 for the VO73 and SO99 dust: hence we may anticipate the strongest red signals to be triggered by VO73 dust, and to a lesser extent SO99 dust. In contrast, the presence of OPAC or DB17 dust will have much less of an impact. Meanwhile the greatest suppression in the green beam would also be expected to come from the VO73 and especially the

SO99 dust types.

It is therefore clear from Figure 5 that the resultant colour will be highly sensitive to the size distribution and the precise composition of the dust. Since typical pink dust colours arise from an increase in red and a slight decrease in green in the RGB rendering, it seems that VO73 and SO99 dust would tend to produce characteristic pink signals predominantly from bin 3, and to a lesser extent bins 1 and 2. OPAC dust shows quite different behaviour with negligible contrast in the red beam for dust in





bin 3, triggering negligible change in colour. OPAC dust in bins 1 and 2 will suppress red signals and increase the green, and it is this pattern which can dominate the total for the OPAC dust. Given the information that Figure 5 provides, it appears that for a given dust loading and size distribution, the greatest intensity of pink signals in the simulated imagery will be produced
by dusts with VO73 and SO99 refractive indices.

T-matrix calculations can be computationally expensive, especially for $x > 10$ (Dubovik et al., 2002). Larger $x$ values occur at shorter wavelengths and larger particle radii: at a wavelength of $8.7\,\mu$m and an effective radius of $13.8\,\mu$m the scattering parameter has a value of 9.97. Estimating and measuring particle aspect ratios is currently an active area of research: for example, Dubovik et al. (2002) set a range of plausible aspect ratios for typical dust particles between 1.6 and 2.2 based on
retrieval errors, preferably between 1.8 and 2.0. During the Saharan Mineral Dust Experiment (SAMUM) in Morocco in 2006, Kandler et al. (2009) took particle samples and found a median $AR$ value of 1.6 for dust particles with radii greater than 250 nm, a value used also by Otto et al. (2009). Meanwhile during an aircraft campaign as part of the the African Monsoon Multidisciplinary Analyses (AMMA) project, in the same year, Chou et al. (2008) measured a median $AR$ value of 1.7. This range of values implies choosing just a few values of $AR$ of interest for further analysis, given that a distribution of $AR$ values
is not practical computationally.

For spheroidal particles, assumptions as to the particle orientation distribution are of vital importance to the resultant extinction properties of the dust. Most authors of previous studies on the light scattering by spheroidal particles (e.g. Dubovik et al., 2002; Wiegner et al., 2009) have assumed the particle orientation to be random, i.e. uniformly distributed, given the paucity of available measurements to make any other assumption. However from a theoretical perspective (Klett, 1995), there is the po-
tential for particles suspended in a stable atmosphere to settle into a horizontally oriented equilibrium with the sides of greatest surface area pointing towards the ground and to space (oblate spheroids with the axis of rotation aligned vertically), a scenario which is more likely for larger particles than smaller ones. The desert atmosphere is typically far from being stable however, so this orientation may not be particularly probable, but it can be argued that this possibility lies at one end of the spectrum of possible orientation distributions. At the other end, Ulanowski et al. (2007) argue that dust particles may be preferentially
aligned vertically, due to electrical charging of the particles. Attempts to measure dust particle orientations are an ongoing area of research (Geier and Arienti, 2014).

Figure 6 describes how concerned we should be by variability in the choice of aspect ratio and the particle orientation, by considering the extinction, absorption and scattering efficiencies for the three IR channels of interest, for the VO73 and SO99 dusts, as a function of aspect ratio. This is carried out for the third COSMO-MUSCAT dust size bin (radii between
0.9 and $2.6\,\mu$m, $r_{\mathrm{eff}} = 1.51\,\mu$m), which Figure 5 indicates gives the highest average extinction values. The calculations for the horizontally oriented dust particles are marked by the solid lines, while randomly oriented dust particles are represented by the dashed lines in panel (a), and it is clear that there is a negligible difference in the extinction properties between spheres and highly elongated but randomly oriented spheroids. In comparison, the extinction is highly responsive to the increasing asphericity of horizontally oriented spheroids, which may be regarded as the upper bound on the light extinction properties for
a given set of particles with a defined size and shape. In contrast vertically aligned particles lead to a reduction in the extinction with increased asphericity, and may be regarded as the lower bound. As an example of the consequences of orientation, the





VO73 dust in bin three at 10.8 $\mu$m gives an extinction efficiency of 1.30 for spherical dust, while for $AR = 2$ spheroidal particles the efficiencies are 1.71 and 0.97 for horizontally and vertically aligned dust, respectively. Hence as a sensitivity study we shall explore further the consequences of various shapes of horizontally aligned particles on the IR imagery in Section 5.3.

For the horizontally oriented dust, the efficiencies at 10.8 and 12.0 $\mu$m for the VO73 dust increase by a very similar factor as each other; however given the higher 'initial' spherical value at 10.8 $\mu$m, this channel increases by a larger magnitude, implying increases in red signal for a given dust loading as the assumed aspect ratio increases. Recall that it is the value of the difference in the efficiencies between channels which determines the change in colour for a given dust loading. Similarly the green signal (marked by the difference in the 10.8 and 8.7 $\mu$m channels) will decrease by more as the aspect ratio increases.

This is especially true for the SO99 dust whose green values will decrease much more with increasing aspect ratio, due to the unusual decrease in efficiency with aspect ratio, a consequence of the real part of the refractive index at 8.7 $\mu$m being less than 1 for this channel and dust type. Considering the ratio between the efficiencies at $AR = 2$ and $AR = 1$ for each dust type and wavelength, there is a very high correlation ($> 0.98$ for the extinction) between this ratio and the real part of the refractive index. Both parts of the refractive index are therefore significant in calculating the extinction properties of spheroidal particles.

Discriminating between the absorption and scattering components (Figure 6(b, c)), it is apparent that both absorption and scattering are significant contributors to the extinction, for all three channels and both dust types. Absorption accounts for between 61 and 78 % of the extinction by spherical dust ($AR = 1.0$) for this set of three wavelengths and two dust types. At 10.8 $\mu$m it is the VO73 dust which is the more absorbing, while the SO99 dust is the more scattering of the dust types; the total extinction is comparable between the two. The differences in the extinction between the two dust types at 12.0 $\mu$m are

governed by the higher absorption of the SO99 dust since the scattering efficiencies in this channel are almost identical to each other. At 8.7 $\mu$m the absorption is greater than the scattering for both dust types, and scattering contributes the least to the overall extinction. It is also clear that both the absorption and the scattering vary with aspect ratio in relation to the real part of the refractive index.

## 4   Comparisons between COSMO-MUSCAT simulated AODs and retrievals of dust AOD

Before investigating the capabilities of COSMO-MUSCAT-RTTOV in simulating brightness temperatures and colours, it is important to assess the relative performance of the COSMO-MUSCAT simulations and the SEVIRI retrievals in estimating the dust loading over the desert regions for the Junes and Julys of 2011-2013. It is important to note that, given the uncertainties in the modelled desert soil properties and the wind patterns, there is no expectation that the model will simulate all the details of the observed dust events. Of more interest is the capacity of the model to simulate a similar range of dust loadings as is

present in the real atmosphere. As an independent source of ground-based AOD data, sunphotometer data from four Saharan AERONET sites (Holben et al., 1998) are considered for validation. Two of these are Fennec AERONET sites in the central desert, Bordj Badji Mokhtar (BBM) in Algeria and Zouerat in Mauritania (Marsham et al., 2013; Todd et al., 2013), active during the summers of 2011 and 2012. The other two are in more upland areas of Algeria and Morocco, Tamanrasset INM (e.g. Guirado et al., 2014) and Ouarzazate, which are also active in 2013. These are summarised in Table 2. As a more direct





measurement of light extinction in the presence of aerosols, it is to be expected that the AERONET AODs more accurately reflect the dust loading than do the SEVIRI retrievals, dependent as the latter are on assumptions as to the pristine-sky brightness temperatures, dust IR optical properties, dust height, etc. Holben et al. (1998) state AERONET AOD uncertainties of the order

of 0.01-0.02 for a new sunphotometer under cloud-free conditions; in contrast, Banks et al. (2013) noted a SEVIRI AOD bias of +0.11 with respect to AERONET measurements from nine North and West African sites in June 2011, with a root-mean-square difference (RMSD) of 0.39. In this comparison AERONET AODs are acquired at 550 nm by using the AOD at 675 nm, scaled to 550 nm using the derived Ångström coefficient (e.g. Eck et al., 1999). Meanwhile the SEVIRI AODs are re-gridded from the pixel scale to the COSMO-MUSCAT 28 km grid, for consistency with the model output.

SEVIRI AODs show better correlation statistics with AERONET than does COSMO-MUSCAT, as shown in Table 3. This is a simple consequence of the fact that SEVIRI and AERONET are measuring the same dust events that have occurred, without the temporal and spatial offsets that they both have with respect to the COSMO-MUSCAT simulations, which have different spatial and temporal distributions. As has been noted previously (Banks et al., 2013), the SEVIRI IR land dust AOD retrieval has a tendency to overestimate the dust loading with respect to AERONET, especially apparent here over the more mountainous

sites at Ouarzazate and Tamanrasset.

Dust activity is at its most intense over BBM, in the central Sahara at the confluence of dust storms produced in Niger, Mali, and southern Algeria, often caused in summer by cold pool outflow events which can trigger haboobs (Farquharson, 1937). The SEVIRI AOD retrievals are well matched here with AERONET, with a positive bias of just +0.03, although the RMSD of 0.38 does imply a degree of variability in their agreement. Meanwhile COSMO-MUSCAT also has a small persistent bias

against the AERONET AODs, of -0.10, however the RMSD is much higher, at 0.93. This implies that large dust events are present in both the AERONET measurements and the COSMO-MUSCAT simulations, but at different times. Figure 7 shows a time-series of the retrieved and simulated AODs over BBM and Zouerat during the Junes (and Julys for Zouerat) of 2011 and 2012, indicating a number of discrepancies in the timing of the dust events. For example, in June 2012 the AODs exceed 3 on two days, on 12th June in the COSMO-MUSCAT simulation, and on 16th June in the AERONET measurements Figure 7(a).

On neither day did the other observe or simulate such a significant event. On the other hand, there are concurrent spikes in dust activity between the datasets on other days, such as in June 2012 on 8th at BBM and 26th at Zouerat.

## 5  Ability to simulate the colour in Desert Dust imagery

### 5.1  Simulated imagery

For initial context of the imagery produced by COSMO-MUSCAT-RTTOV, Figure 8 presents example simulated instantaneous imagery from the pristine-sky and the dust simulations. These images represent simulations of the timeslot in June 2013 of the SEVIRI image in Figure 1, in which a number of related dust events are occurring across southern Algeria and neighbouring regions, with simulated COSMO-MUSCAT AODs (at 550 nm) exceeding 4 on the Algerian/Malian border in the vicinity of BBM. The pristine-sky case in panel (a) is included as a reference image. Differences in dust location and thickness are

5  apparent, as are differences in dust colour due to its assumed optical properties and particle shapes. On a purely visual level,




the VO73 dust simulations appear better able to capture the characteristic pink dust colours observed by SEVIRI at this time than do the SO99 dust simulations, however the depth of the purple colours in the SO99 imagery, while rare, is a possible dust colour that can be seen at other times in the SEVIRI imagery.

Panels (c-f) of Figure 8 show that, qualitatively for this timeslot, horizontally aligned non-spherical particles produce deeper pink (for VO73) and purple (for SO99) colours in the imagery, representing the upper bound on how much influence such a particle shape could have on the IR imagery. For a scene in which a large area of the southern Algerian border regions are covered by dust with an AOD greater than 3, the weak pink colours observed in the spherical dust case in Figure 8(c, d) appear to be insufficient. For reference, the SEVIRI retrieved AODs of the apparent pink dust just to the north of the Algerian/Malian border in Figure 1 vary between $\sim 1.6$ and 2.5. Making the assumption that the particles are non-spherical and aligned horizontally produces more plausible deep pink and purple colours.

## 5.2 Colour patterns in relation to AOD and surface thermal emissivity

A quantitative analysis of the ability of the COSMO-MUSCAT-RTTOV simulations and the dust optical properties' capabilities at reproducing the dust signals and colours present in the SEVIRI observational imagery could be obtained by referencing the simulated brightness temperatures against SEVIRI measurements and retrievals. A direct temporally and spatially colocated comparison would perhaps be an unfair comparison however, since it would stack the uncertainties in the dust spatial and temporal distribution from COSMO-MUSCAT together with both the uncertainties in the dust radiative properties derived from the T-matrix calculations and the uncertainties in the RTTOV simulations. Temporally and spatially co-located observation and model points are not necessarily representative of the same dust storm situation. As an example, the brightness temperature correlations between the SEVIRI measurements and the COSMO-MUSCAT-RTTOV simulations at the lower AOD site of Zouerat vary between 0.89 and 0.92 for the various channels and dust types and shapes, significantly higher than at BBM, where the correlations vary between 0.50 and 0.63. Therefore the variability in the dust loading, and in other properties such as the surface emissivity or the atmospheric moisture, appear to be bigger factors in the brightness temperature agreement, and hence the colour agreement, between the simulations and SEVIRI than is the nature of the dust itself. The differences in dust presence in the simulations and in the observations appear to be obscuring the ability to resolve the differences produced by different assumptions about the dust types themselves.

It is more informative instead to consider comparisons between the SEVIRI observations/retrievals and the RTTOV simulations when they are under the same conditions, which implies comparing points sub-divided within binned variable values such as the dust AOD and surface properties (e.g. thermal emissivity), as well as the dust type and shape. How do these affect the final colour? It is important also to note that such an analysis would be a comparison, not a validation, since the SEVIRI AODs are themselves a derived product based on retrieval assumptions as to dust type (using the VO73 dust refractive indices and spherical particles) and height, atmospheric conditions and surface properties, and have their own uncertainties (Brindley and Russell, 2009; Banks and Brindley, 2013; Banks et al., 2013). Hence comparing the brightness temperatures and colours will also be a useful exercise in understanding the behaviour of the SEVIRI retrievals themselves.





In order to investigate the effects of dust loading and surface emissivity on the resultant brightness temperatures and colours, Figure 9 plots the mean brightness temperatures and colours within specified AOD (at 550 nm) and surface emissivity ranges. This data subset includes successful retrievals and spherical dust simulations from all six months and includes grid cells from across the entire Saharan domain, subdivided into three regimes of the surface emissivity at 8.7 μm. With the lowest emissivities of 0.7-0.8, panels (a, b) are most representative of sandy desert, whereas the high emissivities in panels (e, f) include the most vegetated and/or mountainous areas. The datasets are not co-located in time and space, except inasmuch as they are in the same period of time and have the same bounding domain and resolution, so there are a different number of points which go into the SEVIRI averaged values as compared to the simulation values. Included also in this analysis are the simulations using the OPAC dust refractive indices to show the effects of the extinction values (which form the AODs presented in Figure 5) from another widely used dust type. The left column presents bar charts of the brightness temperatures by channel, for AODs between 0 and 0.2 (left set of bars per channel), and for AODs between 2 and 3 (right), i.e. for near-pristine and very dusty conditions. Meanwhile the right column of panels reinterprets Figure 2 to indicate the averaged colour response in the measurements and simulations to the presence of dust. These plots are similar in concept to those previously presented by Brindley et al. (2012). The lines indicate the progression in AOD for each individual retrieval or simulation dataset sample, with the red and the purple symbols being explored in more detail in the brightness temperature bar charts.

At the lowest AODs, marked by the left set of $T_B$ bars and by the red colour points with AODs of $< 0.2$, the measurements/retrievals and the simulations are most dependent on the surface properties, and tend to have their hottest $T_B$ values. The $T_B$ values of the simulations are very similar to each other, as is to be expected given how little dust is present. It is readily apparent that the simulated $T_B$ values are persistently lower than the SEVIRI measurements: for the three panels the VO73 simulated offset with respect to SEVIRI is between -4.8 and -5.2 K at 8.7 μm, between -7.3 and -7.8 K at 10.8 μm, and between -6.5 and -7.4 K at 12.0 μm. The clear implication of this is that the simulated skin temperatures produced by COSMO-MUSCAT are persistently negatively biased, one potential cause of which may be high-biased soil moisture.

The 'pristine-sky' colours generally lie at the highest green values and lowest red values within their respective panels and AOD-progression lines. Again, the values of the simulated colours at the lowest AODs are almost identical to each other. However together they are offset from the SEVIRI points. The green values are very sensitive to the surface emissivity, with $T_{B087}$ values increasing with increasing emissivity: in the SEVIRI measurements the green colours range from 0.93 in the low emissivity range to 0.66 at the highest emissivities, while in the VO73 simulations this range is from 0.84 to 0.53. The simulation values are persistently negatively offset in the green compared to the SEVIRI measurements, implying perhaps that the 8.7 μm emissivities in the simulations (Figure 3) may be too high. Meanwhile the red values are less sensitive to the emissivity, ranging from 0.53 to 0.36 in the SEVIRI measurements, and from 0.58 to 0.49 in the VO73 simulations. That the red values vary at all with the emissivity at 8.7 μm, when the brightness temperature in this channel is not used in the red derivation, is more indicative of other factors co-varying with the emissivity: the higher emissivities in panels (e-f) of Figure 9 are most associated with moister and more vegetated areas of the Sahel and the more inhabited coastal areas, while also being associated with mountain regions which may be expected to have cooler surface temperatures and will have shallower





atmospheres. Hence it is to be expected that skin temperatures and surface/atmospheric humidities have different characteristics
in the different emissivity ranges.

The overall pattern with increasing AOD is clearly for the colour to become pinker, especially for the SEVIRI measurements
and the VO73 dust simulations (but with the OPAC dust as the exception), increasing the red and decreasing the green. The
VO73 simulations closely follow the same direction of the colour-track as SEVIRI. However what becomes apparent when
looking at the length of the lines is how the change in colour in the SEVIRI retrievals is notably stronger than in the simulations,
in both the red and the green beams. At the highest dust loadings the mean SEVIRI measurements reach red values of $> 0.95$
and low green values of $< 0.24$, whereas the simulated colours do not reach such high red values. This is a consequence of the
differences between the black (SEVIRI) $T_B$ bars from low to high AOD, whereby the measured $T_{B108}$ values drop by more than
do the other channels, by $\sim 15 \, K$ at $10.8 \, \mu m$ and by $\sim 12 \, K$ at $12.0 \, \mu m$, boosting the red colours and minimising the green.
Meanwhile the simulated $T_B$ values drop by much less, and in some cases actually increase, so the colour contrast between
the low and high AOD regimes is correspondingly lower in the simulations. This implies that the COSMO-MUSCAT-RTTOV
dust simulations may be warm-biased, perhaps because of insufficient dust extinction, a warm-biased model atmosphere, or a
low-biased model dust vertical distribution.

The $T_{B120}$-$T_{B108}$ difference governs the strength of the red beam, and explains why the VO73 dust simulations are most
similar to the SEVIRI measurements/retrievals. For AODs between 2 and 3 the mean $T_{B120}$-$T_{B108}$ difference over the lowest
emissivity surfaces is $+2.08 \, K$ in the SEVIRI measurements, $+0.43 \, K$ with the VO73 dust, $-0.37 \, K$ with the SO99 dust, and
$-2.12 \, K$ with the OPAC dust (less than in the 'pristine' case). The $T_{B120}$ values produced by the SO99 dust reduce by more
than those produced with the VO73 dust, and also the $T_{B108}$ values are not reduced as much by the SO99 dust compared to the
VO73 dust. The combination of these factors encourages more red response to dust with the VO73 compared to the SO99 dust,
and gives rise to redder, and therefore pinker, dust imagery with the VO73 dust compared to the other dust types. Meanwhile
the SO99 dust maintains higher $T_{B087}$ values than do the other dust types, resulting in typically lower green values: hence the
SO99 dust follows a more vertical track on the red-green plots than does the VO73 dust, following a path through more purple
colours.

The OPAC dust is the exception to the principle that dust reddens the imagery, with a colour-track which instead heads left
to lower red values with increasing AOD, while also reducing in the green values over the lower emissivity regions. Increasing
levels of OPAC dust have the consequence of reducing the pinkness of the imagery and instead producing deeper blue colours.
This is widely divergent from what is observed in the imagery, and is a consequence of the very high imaginary part of the
refractive index in the $12 \, \mu m$ channel (Figure 4(b)), leading to high values of the extinction in this channel (Figure 5), and
hence OPAC dust has a tendency to produce higher $T_{B108}$ than $T_{B120}$ values.

From the change in colour it is apparent at what dust loadings the scenes start to turn pink, which is different for each dust
type and emissivity domain. At the lowest AODs of $< 1.5$ (red, orange, yellow, and green symbols) the SEVIRI points have a
lighter blue appearance than do the simulations, in all three emissivity ranges. Note that there is a high degree of spread in the
red beam for the AODs between 1 and 1.5 (in panel (b) these are 0.24 for SEVIRI, 0.16 for VO73): therefore in this AOD range
it is possible both in the SEVIRI and the VO73 imagery for a dust event to appear in colours ranging from light pink (red $> 0.8$)

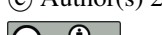



to deep blue over sandy desert surfaces (panel (b)). Over more mountainous and/or vegetated regions (panel (f)), the VO73

simulations ($1 < AOD < 1.5$) exhibit similar red values as those seen over sandy desert surfaces, giving a similar possibility for pink colour values. This is not true of SEVIRI, whose mean red values are very much lower than over sandy surfaces (0.31 as opposed to 0.57), and hence pink colours are highly unlikely in this AOD range. Over high emissivity surfaces, SEVIRI pink colours are still unlikely at AODs $> 1.5$ and $< 2$ (blue symbols), where the mean red value is just 0.41. Meanwhile in this AOD range the SEVIRI points in panel (b) come into alignment with the VO73 simulations, and at this stage both the SEVIRI

measurements and the VO73 and SO99 simulations have started moving into the more purple/pink colour domain. Finally for the dustiest cases at AODs $> 2$ (purple symbols), the SEVIRI measurements move decisively ahead to become markedly redder than all of the simulations in all emissivity ranges, and less green than the VO73 simulations.

    This colour progression tells us more about the behaviour of the SEVIRI AOD retrieval, in that SEVIRI retrieved AODs $< 1$ are unlikely to be frequently noticeable by their 'pinkness' in the Desert Dust imagery, while at AODs $> 2$ the imagery is very

clearly pink. Between AODs of 1 and 2 is the transition phase, whereby dust may or may not be likely to be visible in the imagery, depending on the background conditions: over low emissivity surfaces the scenes may start to appear subtly pink between AODs of 1 and 1.5, whereas over high emissivity surfaces the scenes are likely to remain blue or faintly purple until AODs of $\sim 2$. There is a substantial step-change in the image pinkness in the SEVIRI retrievals at high emissivities, while at lower emissivities there is a much smoother progression with AOD. The VO73 and SO99 simulations show similarly smooth,

relatively linear responses to AOD, and start to show the potential for subtle pink or purple colours at AODs between 1 and 2.

    This step-change in the behaviour of the SEVIRI colours highlights the role of water vapour as another important contributor to the overall image colour. As noted in Section 2.1, atmospheres with high moisture content will have a tendency to inhibit the red beam in the imagery, giving rise to bluer overall colours. The high emissivity surfaces encompass moist atmospheric regions such as over the Sahel, and also mountain areas which can be slightly drier due to the shortness of the atmospheric

column. Considering the points in panel (f), between AODs of 0 and 0.2 the mean ERA-Interim and COSMO-MUSCAT column moistures are 18.4 and 15.6 mm, both similarly dry values and hence the resultant colour is predominantly a function of the properties of the background surface. Between AODs of 1 and 1.5 (green symbols), these moisture values are 32.9 and 27.7 mm: the ERA-Interim values are particularly moist between AODs of 1 and 2, explaining why despite the increase in AOD the mean SEVIRI red colours are still restricted to values of $\sim 0.4$ in this range. In contrast the drier COSMO-MUSCAT values

do not restrict the red colours at these AODs to the same extent. Finally at the highest AODs ($> 3$) the mean ERA-Interim and COSMO-MUSCAT column moistures are 24.2 and 33.5 mm, a reversal of the previous situation at intermediate AODs, helping to explain why the COSMO-MUSCAT-RTTOV red colours tend not to reach as high values as do the SEVIRI measurements when the retrieved or simulated dust loading is very high.

## 5.3   Sensitivity to particle shape assumptions

A sensitivity study exploring the implications of the choice of particle aspect ratio is presented in Figure 10, for horizontally aligned spheroidal particles. This should be regarded as the upper bound on the light extinguishing properties of the available dust fields. Figure 10 adapts the left column of Figure 9 to analyse the mean brightness temperatures and colours within the



emissivity ranges as a function of aspect ratio for the VO73 dust, for dusty scenes again with COSMO-MUSCAT and SEVIRI

AODs between 2 and 3. Following Chou et al. (2008), the value of 1.7 is tested here, and a value of 2.0 is also considered in order to test the sensitivity of the simulations to even higher aspect ratio values. The persistent warm offset in the dusty simulations reoccurs here, although with higher aspect ratios the brightness temperatures cool slightly in the direction of the SEVIRI measurements/retrievals. This cooling with increased aspect ratio is most readily apparent in the $10.8\,\mu$m channel, consistent with the gradient of the extinction efficiencies of this channel with respect to aspect ratio displayed in Figure 6.

A greater reduction in the $10.8\,\mu$m channel compared to the other channels implies a boost to the red channel and a greater reduction in the green channel, and indeed this is clearly the case for the colour values in the right column. Approximately half of the red colour discrepancy between the SEVIRI retrieval-measurements and the spherical VO73 dust is removed if the dust is assumed to have an aspect ratio of 2. The green beam is reduced, but it is interesting to note that the fractional reduction in the green with respect to the SEVIRI retrieval-measurements is greatest over the highest emissivity surfaces in panel (f),

where the increased extinction of the spheroidal dust is more effective at masking the background surface signal than that of the spherical dust.

VO73 dust particles with an aspect ratio of 2 do not fully agree with the depth of the red beam and the weakness of the green beam that gives rise to the intensity of pinkness that is noticeable in the SEVIRI imagery (when SEVIRI AODs are used). This suggests that, even with the highly favourable assumption of horizontally oriented dust, the particles should be more aspherical

still, with yet higher aspect ratios. However given the distribution of measured values in the literature, varying between 1.6 and 2.2 (Chou et al., 2008; Dubovik et al., 2002; Kandler et al., 2009) as outer bounds of the spheroidal shape distribution, it is dubious that such elongated spheroidal particles exist in such quantities as to justify such an assertion. More plausible is that the assumption of spheroidal particles is itself insufficient to describe the full range of dust particle shapes present in the Saharan atmosphere. The spheroidal assumption is a step up in complexity compared to the assumption of spherical particles,

but it is clear that there is a much wider range of irregular shapes present within the real atmosphere (e.g. Muñoz et al., 2007).

What all of these plots show is that there are systematic offsets between the dust simulated colours and the SEVIRI measurements even after spatial and temporal mismatches have been accounted for, and that there are numerous variables (dust loading, type and shape, surface emissivity and atmospheric moisture, for example) which act as controls on the resultant colours.

## 6  Conclusions

A new imaging and analysis tool has been established in order to simulate SEVIRI Desert Dust RGB imagery over the Sahara in the infrared, by combining three-dimensional dust concentration output from the COSMO-MUSCAT regional aerosol transport model with radiative transfer simulations using the RTTOV program. This paper describes the proof-of-concept for the model design, and presents initial results of the understanding that may be gained through the analysis of this output. Comparisons are performed with SEVIRI measurements and AOD retrievals, as well as over AERONET sites, making use of the ground-based AOD data. COSMO-MUSCAT-RTTOV has been shown to have some skill at reproducing the characteristic pink colours observable in the imagery, although it is clear that there are systematic differences between the simulations and the observations,




with numerous influences on these differences arising from our incomplete understanding of the nature of the dust itself, as

well as of the background surface and the atmospheric state.

Dust is assumed to be spherical or spheroidal in shape in the simulations, and it is clear from sensitivity studies enabled by COSMO-MUSCAT-RTTOV that an assumption of spherical dust may not be sufficient for understanding the radiative properties of dust in the infrared, and hence for simulating the presence of dust in the SEVIRI imagery. This is consistent with previous studies which have identified the assumption of spherical dust as a major source of error in measuring radiances at the

top-of-the-atmosphere (e.g. Kahnert et al., 2005). In general, the more spheroidal the dust, the greater the extinction properties and the greater is the influence of the dust on the simulated brightness temperatures and colours, but only if the dust is assumed to have some degree of horizontal alignment. The case of randomly oriented elongated spheroids shows comparatively very little deviation from the spherical case. Hence the IR extinction properties of spheroidal dust are highly dependent on their orientation, and they only exhibit a marked increase in values compared to the spherical case when they are assumed to be

aligned horizontally.

The influence of wavelength-resolved IR refractive indices has been tested: of a range of dust refractive index databases considered, only the VO73 (Volz, 1973) and SO99 (Sokolik and Toon, 1999; Helmert et al., 2007) dust types cause sufficient extinction in the three IR channels to approximate the characteristic pink dust colours observed in the imagery. An assumption of OPAC dust would have an implausible effect on the colour of the imagery, corresponding to an increase in blue colours with

increasing AODs, due to its particularly strong extinction properties at $12.0\,\mu$m.

The dust particle size distribution is important to the colour, and its significance to the colour change is controlled by the inter-channel differences in extinction. Figure 5 shows that for larger particles in bin 5 (radii between 7.9 and $24\,\mu$m), there is a considerable flattening in the contrast between the extinctions at 10.8 and at $12.0\,\mu$m as compared to the extinctions in bins 1-3 (radii between 0.1 and $2.6\,\mu$m). Recent in situ measurements (e.g. Ryder et al., 2013) have implied that there is a bigger

coarse mode in lofted dust above the Sahara than had previously been measured, however these T-matrix simulations imply that size distributions dominated by coarse mode particles would result in a weak response of the colour in the IR imagery. Even if the simulated mass concentration in the largest bin were three to four orders of magnitude higher than they are currently simulated to be, the contrast between channels in this bin is not conducive to bolstering distinctive pink colours in the imagery. At least with respect to the COSMO-MUSCAT-RTTOV simulations, it is the dust in bin 3 with $r_{\mathrm{eff}} = 1.5\,\mu$m which produces

the greatest response in the dust colour in the imagery. This has consequences for the distinctiveness of dust in the IR imagery, for example a lofted dust plume with size distributions weighted towards smaller dust particles may be more apparent in the imagery than a near-source dust plume with a greater fraction of coarse particles.

Comparisons with SEVIRI measurements and retrievals show that the simulations have a tendency not to reproduce the full depth of colour that can be observed from SEVIRI. The deepest pink colours are produced by maximising the red beam and minimising the green beam in the imagery, by reducing the values of $T_{\mathrm{B108}}$ with respect to both the values of $T_{\mathrm{B087}}$ and $T_{\mathrm{B120}}$. The simulated brightness temperatures are often low-biased compared to the measured values when the dust loading is low, and high-biased when the dust loading is high, however it is the contrast between the spectral channels which governs the dust influence on colour. Despite the skill of the VO73 dust simulations in producing pink colours, these simulations still



have a tendency to under-predict the red beam and over-predict the green beam under conditions of high dust loading, leading
to a smaller simulated colour response to the presence of dust in the simulations than in the observations. This appears at
least in part to be a result of higher total column water vapour values simulated by COSMO-MUSCAT for points co-located
with the strongest dust events. As for the SEVIRI retrievals and observations, these exhibit a step-change behaviour over
higher emissivity surfaces and across particularly moist atmospheres (column moistures $\gtrsim 25\text{-}30\,\text{mm}$), whereby the pinkness
characteristic of dust is only substantially visible in the imagery when the retrieved AOD is $\gtrsim 2$. Over the lower emissivity
surfaces characteristic of sandy deserts, where the atmosphere is on average rather drier, SEVIRI AOD retrievals indicate that
pink dust is observable in the imagery at a lower AOD threshold of between 1 and 1.5.

In a follow-up paper, we will consider the influences of other dust, surface, and atmospheric factors, in order to assess
their relative contributions to the resultant colour. For example, dust height is a key variable for IR measurements/simulations
of dust, since this governs the temperature of the lofted dust layer. This paper has shown that atmospheric moisture has the
potential to 'hide' dust in the IR imagery (e.g. Brindley et al., 2012), this is also a control on dust colour still to be assessed in
detail. The characteristics of the dust itself are insufficient to completely describe the final colour in the imagery.

*Data availability.* COSMO-MUSCAT and RTTOV model output data are available on request from the authors. AERONET data are available from the NASA GSFC at http://aeronet.gsfc.nasa.gov/ (Holben et al., 1998). SEVIRI AOD retrieval data over land are curated by Helen
Brindley, of the Physics Department at Imperial College London (Brindley and Russell, 2009). SEVIRI data are available from EUMETSAT,
currently at https://www.eumetsat.int/website/home/Data/index.html. The RTTOV program is available from EUMETSAT's NWP SAF facility, currently at https://nwpsaf.eu/site/software/rttov/. The database of refractive indices produced by Di Biagio et al. (2017) is publicly
available in the supplement to their paper, with the doi:10.5194/acp-17-1901-2017-supplement. The OPAC dataset (Hess et al., 1998; Koepke
et al., 2015) is available at www.rascin.net.

*Competing interests.* The authors declare that they have no conflict of interest.

*Acknowledgements.* Jamie Banks and Kerstin Schepanski acknowledge funding through the Leibniz Association for the project "Dust at the
Interface – modelling and remote sensing". We thank the PIs of the four AERONET sites used in this study for maintaining the sites and for
providing the data: Martin Todd at the University of Sussex for the Bordj Badji Mokhtar and Zouerat sites, and Emilio Cuevas-Agullo at the
Izana Atmospheric Research Center for the Tamanrasset INM and Ouarzazate sites. Deutscher Wetterdienst (DWD) have provided access to
the COSMO model. We also thank Hartwig Deneke from the Leibniz Institute for Tropospheric Research for his productive comments and
ideas at various stages during the course of this work. JRB would like to thank James Hocking at the UK Met Office for his helpful advice in
setting up the RTTOV component to the simulation system. Finally, JRB and BH would also like to thank Jürgen Helmert, now at DWD, for
the provision and preparation of the VO73 and SO99 dust optical properties data (Volz, 1973; Sokolik and Toon, 1999; Helmert et al., 2007).



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



**Table 1.** Size ranges and effective radii of the five COSMO-MUSCAT size bins ($\mu$m). Mineral densities are assumed to be $2.65\,\mathrm{g\,cm^{-3}}$.

| Bin | Radius range | Effective radius |
|-----|-------------|------------------|
| 1 | 0.1 - 0.3 | 0.169 |
| 2 | 0.3 - 0.9 | 0.501 |
| 3 | 0.9 - 2.6 | 1.514 |
| 4 | 2.6 - 7.9 | 4.570 |
| 5 | 7.9 - 24.0 | 13.800 |

**Table 2.** AERONET sites of interest with associated latitudes and longitudes (in °N and °E), altitudes (m), June emissivities for the MSG-3 SEVIRI 8.7 $\mu$m channel, and periods of available data for the summers between June 2011 and July 2013.

| Site | Lat. | Lon. | Alt. | Emissivity | Period |
|------|------|------|------|-----------|--------|
| Bordj Badji Mokhtar | 21.38 | 0.92 | 400 | 0.80 | Junes 2011 and 2012 |
| Ouarzazate | 30.93 | -6.91 | 1136 | 0.92 | 2012 and 2013 |
| Tamanrasset INM | 22.79 | 5.53 | 1377 | 0.94 | All |
| Zouerat | 22.75 | -12.48 | 343 | 0.83 | June 2011 - July 2012 |

Zhang, L., Gong, S., Padro, J., and Barrie, L.: A size-segregated particle dry deposition scheme for an atmospheric aerosol module, Atmospheric Environment, 35, 549–560, 2001.



**Table 3.** COSMO-MUSCAT (C-M) and SEVIRI AOD comparisons with AERONET. The mean AODs for the four sites (with numbers of points in the comparisons in parentheses), specified using AERONET data, are: 0.97 at BBM (85), 0.22 at Ouarzazate (213), 0.41 at Tamanrasset INM (250), and 0.55 at Zouerat (229).

| Site | C-M/AER | SEV/AER | C-M/SEV |
|---|---|---|---|
| BBM | | | |
| $r^2$ | 0.22 | 0.86 | 0.13 |
| offset | -0.10 | +0.03 | -0.13 |
| RMSD | 0.93 | 0.38 | 0.92 |
| Ouarzazate | | | |
| $r^2$ | 0.43 | 0.64 | 0.18 |
| offset | -0.10 | +0.26 | -0.36 |
| RMSD | 0.21 | 0.36 | 0.49 |
| Tamanrasset INM | | | |
| $r^2$ | 0.27 | 0.85 | 0.16 |
| offset | +0.18 | +0.14 | +0.04 |
| RMSD | 0.55 | 0.29 | 0.63 |
| Zouerat | | | |
| $r^2$ | 0.58 | 0.74 | 0.33 |
| offset | -0.27 | +0.06 | -0.33 |
| RMSD | 0.40 | 0.26 | 0.48 |

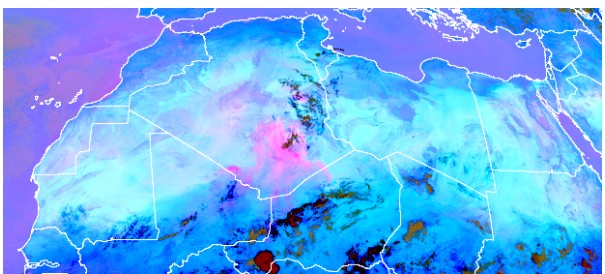

**Figure 1.** Desert Dust RGB image from MSG-3 SEVIRI, 1200 UTC on 25th June 2013.





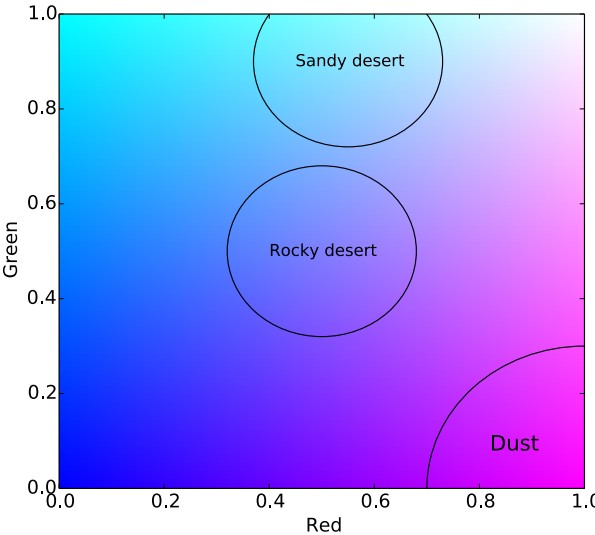

**Figure 2.** RGB colours as a function of red and green beams, with the blue beam fixed to a value 1. Approximate ranges of characteristic daytime colours for dust and two surface types are overlaid.

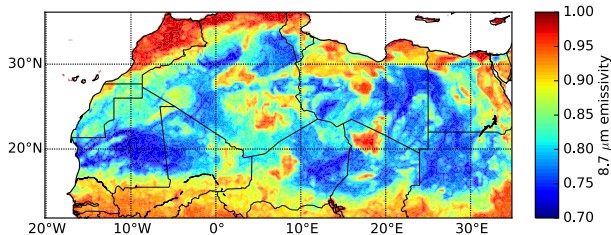

**Figure 3.** Map of climatological surface emissivity for the MSG-3 SEVIRI channel at 8.7 $\mu$m used by the RTTOV program for June, derived by Borbas and Ruston (2010) using MODIS satellite data.





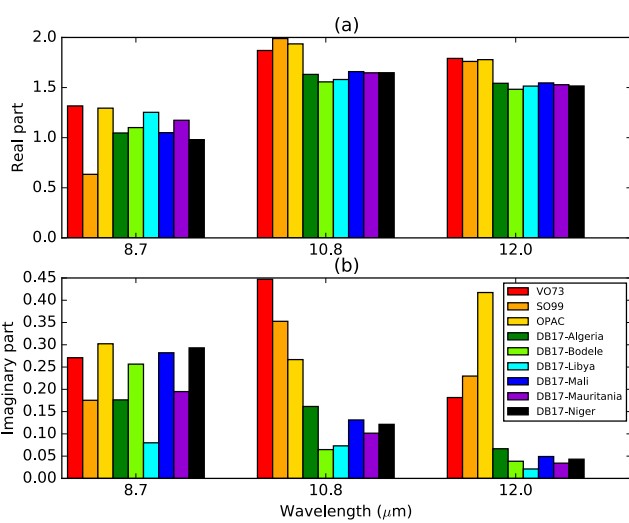

**Figure 4.** Plots of the real (a) and imaginary (b) parts of the dust refractive index, for nine of the dust databases considered in this study, which have been convolved across the MSG-3 SEVIRI filter functions.



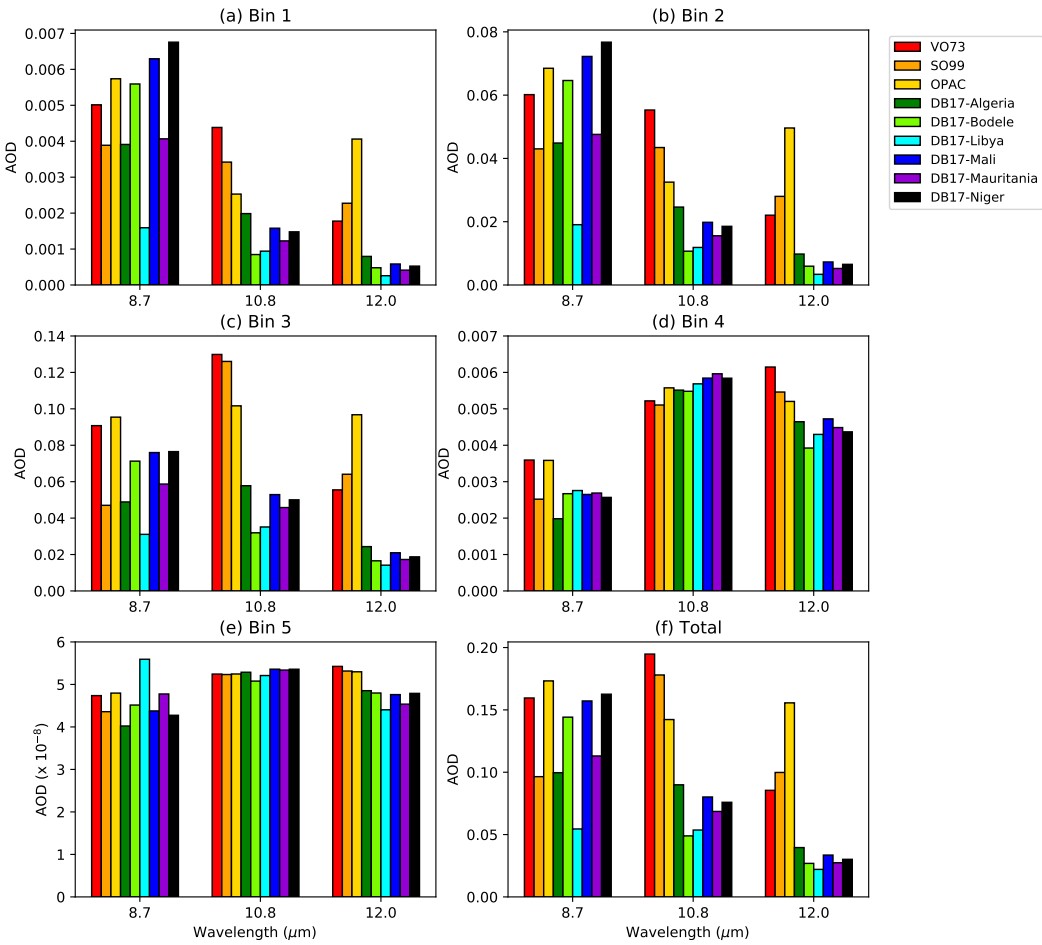

**Figure 5.** Plots of atmospheric column-integrated IR AODs for the five size bins and their total, over the BBM site in southern Algeria (21.38°N, 0.92°E), averaged for 0900, 1200, and 1500 UTC for June and July 2013, for MSG-3, and for each set of IR refractive indices considered in this study. Note that each panel has a different y-axis scale, and note also that each panel is multiplied by one temporally averaged value of the column-integrated number density, so the pattern of AOD within each panel is identical to the pattern of the extinction cross-section for that panel (i.e. for any panel the value of the correlation of the extinction cross-section and the AOD is 1). The particles are spherical.





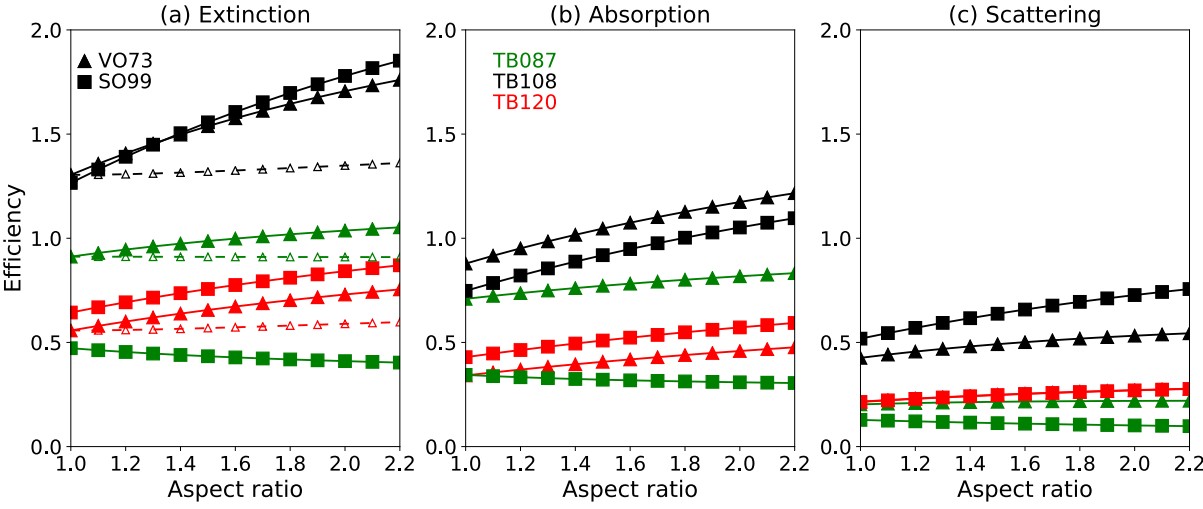

**Figure 6.** Plots of the derived (a) extinction, (b) absorption and (c) scattering efficiencies for the VO73 and SO99 dust types, as a function of particle aspect ratio, for COSMO-MUSCAT dust size bin 3 ($r_{eff} = 1.51\,\mu$m) for the three MSG-3 SEVIRI channels at 8.7, 10.8, and 12.0 $\mu$m. The solid lines are for horizontally aligned particles, the dashed lines for randomly oriented particles. For clarity, the randomly oriented case is only included in the extinctions for VO73 dust.





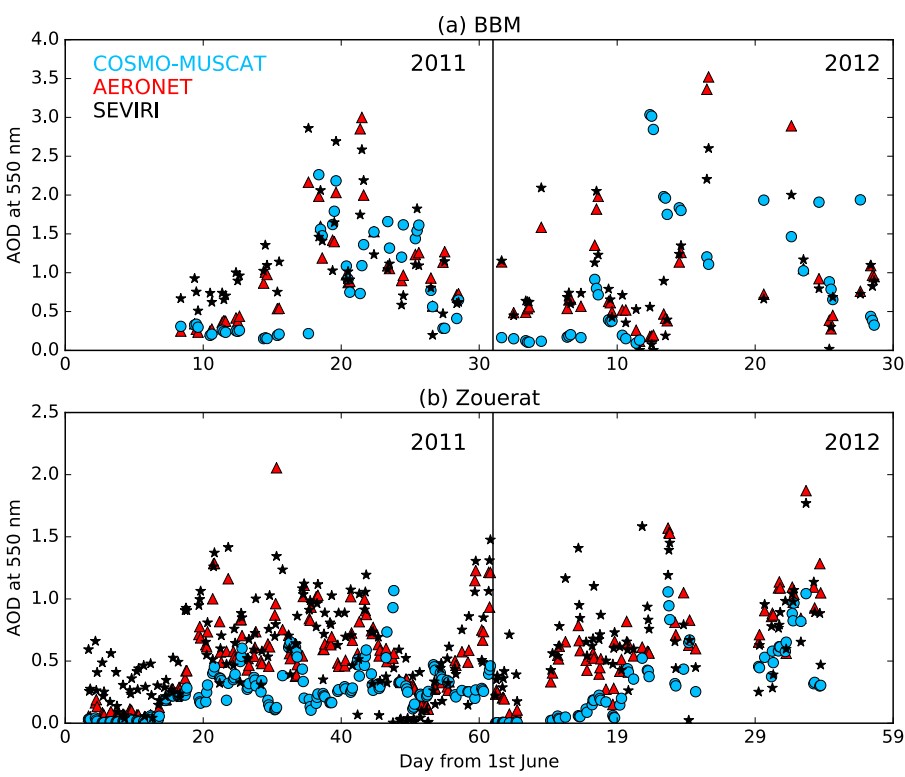

**Figure 7.** Time-series of measurements, retrievals, and simulations of AOD at 550 nm over the Fennec BBM and Zouerat AERONET sites, during 2011 and 2012 when AERONET data are available, linked into one time-series: (a) BBM, the two June months; (b) Zouerat, the four months of Junes and Julys.




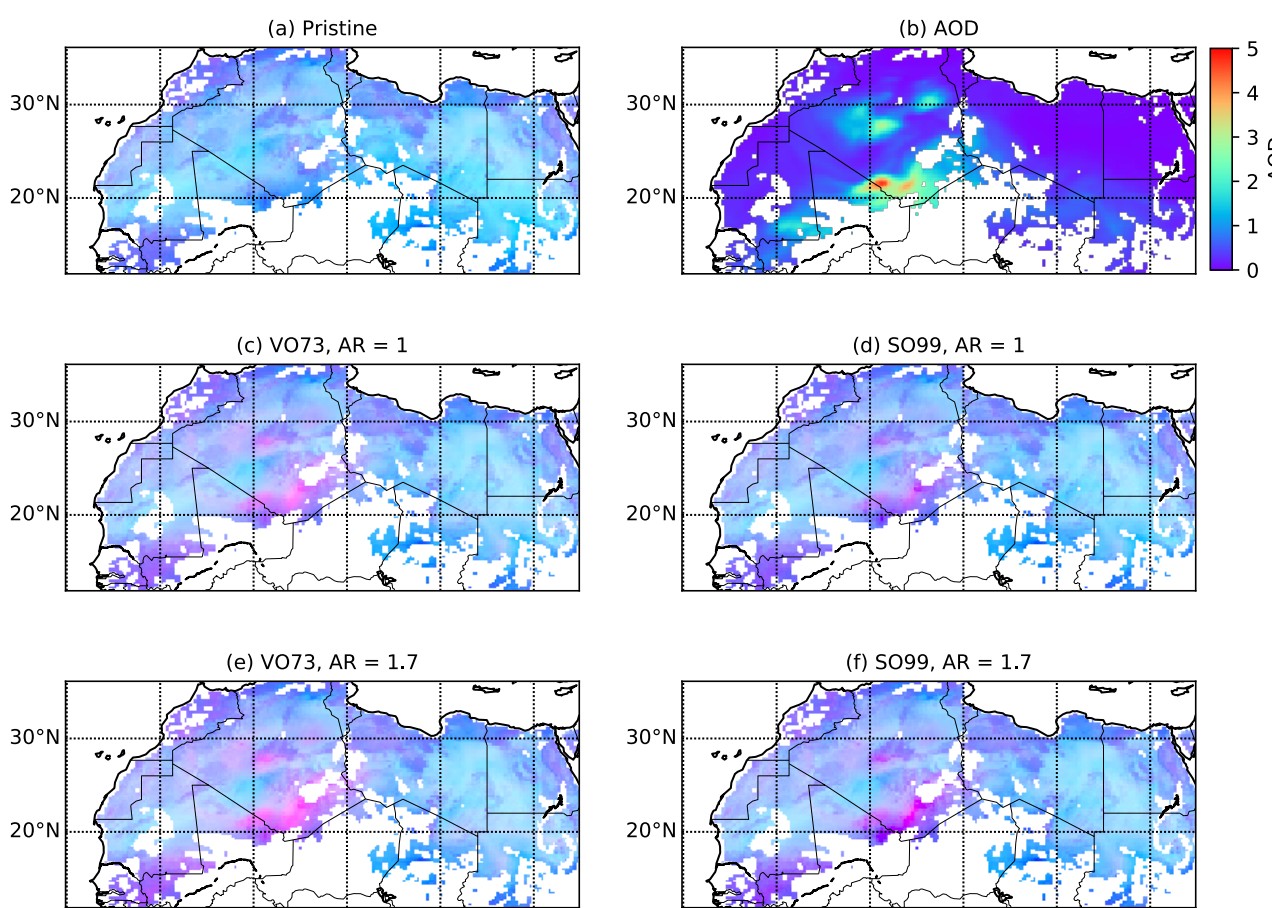

**Figure 8.** Simulated SEVIRI Desert Dust RGB images, 1200 UTC on 25th June 2013. (a) Pristine-sky; (b) COSMO-MUSCAT AOD at 550 nm; (c, e) VO73 dust for $AR$ values of 1 and 1.7 (horizontally aligned); (d, f) the SO99 dust mixture for the two $AR$ values.



**Figure 9.** Mean brightness temperatures and colours across the domain for three ranges of surface emissivity values, for the SEVIRI measurements/retrievals and three of the dust simulations, for the Junes and Julys of 2011-2013. Using emissivity at 8.7 $\mu$m, the panels are: (a, b) emissivity from 0.7-0.8; (c, d) 0.8-0.9; (e, f) 0.9-1.0. The simulated dust is spherical. Per panel of brightness temperatures (left column), within each individual wavelength set there are two groups of bars: the striped bars on the left are points with AODs between 0 and 0.2 (i.e. the most pristine-sky case); the bold bars on the right are points with AODs between 2 and 3. The mean colours (right column) are plotted within specified AOD ranges denoted by the coloured symbols: red is 0-0.2; orange is 0.2-0.5; yellow is 0.5-1; green is 1-1.5; blue is 1.5-2; purple is 2-3; black is > 3. The blue beam is held fixed at 1. Error bars denoting the standard deviations of the mean values are only included, for clarity, for three of the SEVIRI and the VO73 points.





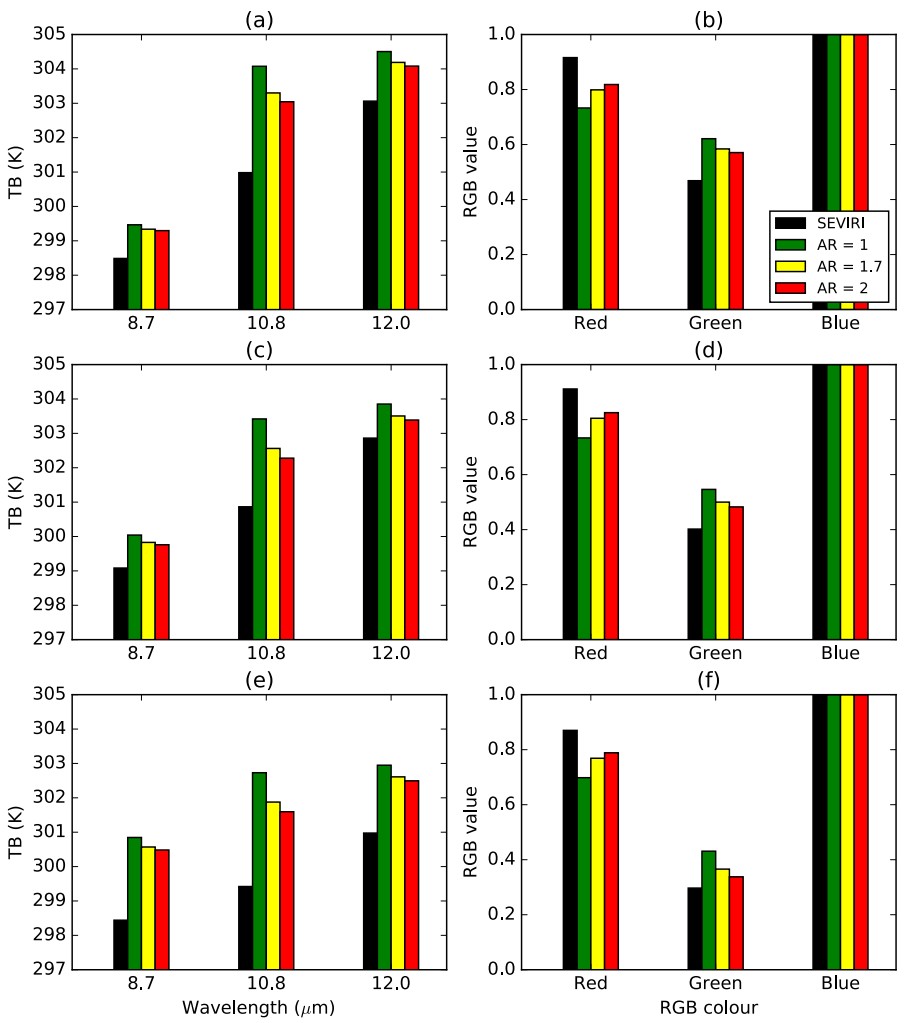

**Figure 10.** Mean brightness temperatures and RGB colours across the domain for the three ranges of surface emissivity values, for the SEVIRI measurements/retrievals and tested for the VO73 dust simulations with the three (horizontally aligned) aspect ratios of 1, 1.7, and 2, for the Junes and Julys of 2011-2013. Only points with AODs between 2 and 3 are included. Using emissivity at 8.7 $\mu$m, the panels are: (a, b) emissivity from 0.7-0.8; (c, d) 0.8-0.9; (e, f) 0.9-1.0.