# Peer review of "The influence of dust optical properties on the colour of simulated MSG-SEVIRI Desert Dust infrared imagery"

_Atmospheric Chemistry and Physics, 2018_

## Referee Comment (RC1) · Anonymous Referee #1 · 3 Apr 2018

General comments:

Overall this is a comprehensive well written examination of how various dust aerosol properties affect long-wavelength IR (LWIR) bands that SEVIRI can measure.

When reading the title I had the impression the paper was going to discuss visible light imagery since these wavelengths actually produce a perception of color. Perhaps using the term "infrared color" or other wording to indicate infrared wavelenghts are the main focus, given that "false" color is being used.

Other specific items:

Page 4 line 2: to clarify, the B beam is just a scaled tb10.8 value while the R and G beams use actual differences between channels.

Page 4 line 11: were only cases chosen where the blue beam equals 1, or were the values normalized somehow?

Page 4 lines 20-30: Would it be possible to label locations of various dust types within figure 8 or another figure?

Page 10 line 27: Does the particle aspect ratio also affect the backscatter fraction, by enhancing the phase function as we approach 180 degrees scattering angle? This does happen at visible wavelengths so I wonder if this also applies with the IR?

Page 11 line 27: Wording could be "It is also important"

Page 18 line 27: With the mention of needed 3-4 more orders of magnitude of coarse dust, is this mainly due to the smaller mass concentration, smaller area to volume ratio or some other factors? It seems the scattering efficient Q should be high enough for coarse particles.

Conclusions:

A good summary and I'm happy to see a mention of cooler elevated dust, and the moisture hiding of dust. Is it possible that consideration of other channels in the shorter IR and visible wavelengths would further help to constrain aerosol properties with satellite imagery? There are papers that talk about both visible and IR that might be considered as references such as this one: Correlated IR spectroscopy and visible light scattering measurements of mineral dust aerosol. Perhaps this can also be highlighted as future work?

https://agupubs.onlinelibrary.wiley.com/doi/full/10.1029/2010JD014389

---

## Referee Comment (RC2) · Anonymous Referee #2 · 21 Apr 2018

First, I sincerely apologize for the delay of my revision.

The paper by Banks et al. analyses the sensitivity of SEVIRI imagery retrievals to dust optical properties and shape based on simulations with the COSMO-MUSCAT model and optical calculations. COSMO-MUSCAT generate the 3-D filed of dust aerosol concentration as a function of the size distribution, then used as input to an optical code to estimate the dust extinction and absorption at different wavelengths. The impact of using different refractive index datasets and different assumptions on the particle shape are evaluated through comparison with the RGB and brightness temperatures retrievals from SEVIRI.

[Figure]

The paper is well written and well organized. The broad context and the specific problematic are described in a clear way, and the methods and objectives clearly identified. The number of figures is appropriate and they well illustrate the main findings.

I have nonetheless some remarks concerning the size and refractive index (RI) assumptions in the study and their impact on the results.

1. The representation of the size distribution of dust and the sensitivity to the coarse fraction is not discussed. In the paper it is shown that the fifth bin above about $8\mu$m in diameter as simulated by COSMO-MUSCAT basically do not contribute to the dust load. I wonder how this compares to field observations of the size distribution in particular from FENNEC close to source regions. Moreover, and based on the comparison with field observations, a sensitivity of the results to the coarse size of dust should be added in the paper.

2. When comparing the different refractive index datasets in Figure 4, it is evident a strong difference at 10.8 and 12$\mu$m between Volz, Sokolik, and OPAC data compared to the new laboratory estimates of the RI. This difference may strongly impact the results of this study. I wonder if this is not due to a bias of some of the datasets at some wavelengths. This is for instance discussed for OPAC at 12$\mu$m due to the large assumed quartz content for this dataset, but I would like a similar discussion also for the strong Volz peak at 10.8$\mu$m. Is this strong peak reasonable or the dataset is biased/shifted in wavelength compared to the other datasets due to differences in the retrieval procedure or other? I suppose this point should be better addressed given that the sensitivity to the refractive index is the core of the paper.

3. On the same line, I would appreciate that all the refractive index datasets are included in Fig. 9. I have the impression looking at this figure that the BT is overestimated at 10.8 $\mu$m in the Volz dataset and I wonder if this is not linked to a possible bias of the Volz dataset at 10.8$\mu$m compared to all the other estimates of the dust RI used in the paper.

Minor comments

The right panel of figure 9 is not very clear to me; please, better explain its content and message.

I agree with the other reviewer in asking to add the word "infrared" in the title.

---

## Author Comment (AC1) · 31 May 2018

**Responses to reviews of "The influence of dust optical properties on the colour of simulated MSG-SEVIRI Desert Dust imagery"**

We thank very much the anonymous referees for their very helpful comments, which have helped us to improve the manuscript. Below are our responses to their comments, and a list of relevant changes. Referees' comments are included in bold, changes to the text in the manuscript are included in italics.

[Figure]

**Referee 1**

**When reading the title I had the impression the paper was going to discuss visible light imagery since these wavelengths actually produce a perception of color. Perhaps using the term "infrared color" or other wording to indicate infrared wavelenghts are the main focus, given that "false" color is being used.**
1) We thoroughly agree that the word 'infrared' belongs in the title. We have placed the word towards the end of the title, just before 'imagery', so that the title is now: "*The influence of dust optical properties on the colour of simulated MSG-SEVIRI Desert Dust infrared imagery*".

**Page 4 line 2: to clarify, the B beam is just a scaled tb10.8 value while the R and G beams use actual differences between channels.**
2) This is true, the blue beam is a scaled brightness temperature value within a specified range, whereas the red and green beams are scaled brightness temperature differences. This sentence has been reworded slightly to clarify this.

**Page 4 line 11: were only cases chosen where the blue beam equals 1, or were the values normalized somehow?**
3) No data are present within Figure 2, this is simply a representation of the colours available as a function of the red and green beams, with the blue beam fixed to its maximum value of 1. However, this is a more pertinent issue with respect to Figure 9 (now Figure 10), which does use observation/retrieval and simulation data. The vast majority of points which go into these comparisons have blue values of 1, but not all do. This subset of data is from the summer daytime cloud-free Sahara, so the apparent surface temperature is almost always greater than the maximum of 289 K in the blue range (Equation 4). Of the 4,227,973 points in the SEVIRI dataset subsample, 493 (i.e. 0.012%) have blue values less than 1. Meanwhile in

the VO73 simulations, 2,311 of the 5,984,864 points (i.e. 0.039%) have blue values less than 1. The minimum SEVIRI blue value is 0.74, while the minimum VO73 blue value is 0.54. We do not screen out those points which have blue values of less than 1. Three extra sentences on this have been added to the third paragraph of Section 5.2.

**Page 4 lines 20-30: Would it be possible to label locations of various dust types within figure 8 or another figure?**
4) Broadly speaking within this work it is not possible to identify dust types by location. The VO73 dust, for example, is a composite dust type analysed from samples taken in Barbados of transported dust, possibly originating from multiple locations within the Sahara. Most dust events are unlikely to correspond precisely to any one dust type, depending on the mixing state. However, we are happy to mark the relevant countries and locations on Figure 1, to provide more geographical context.

**Page 10 line 27: Does the particle aspect ratio also affect the backscatter fraction, by enhancing the phase function as we approach 180 degrees scattering angle? This does happen at visible wavelengths so I wonder if this also applies with the IR?**
5) The aspect ratio does indeed change the backscatter parameter in the IR, actually having a tendency to decrease the backscatter parameter with increasing aspect ratio. This is especially the case for the larger size bins. In the figure on p. C11 we plot the normalised VO73 dust phase functions, for the three MSG-3 SEVIRI channels in the respective columns, and with the five rows indicating the bins from 1-5. The lines are colour-coded by aspect ratio for horizontally-orientated dust: blue is $AR$ = 1, green is $AR$ = 1.7, and red is $AR$ = 2. The resultant back-scatter parameter ($bpr$) for each combination is written in each panel, as are the values of the scattering parameter $x$. The smallest particles have backscatter parameters closest to 0.5, i.e. they display the greatest symmetry, while the largest particles are the most forward-scattering.

The particles in bins 1 and 2 have radii much less than the wavelength, and so more closely approximate the Rayleigh regime. Meanwhile bin 3 has $x$ values approximately equal to 1, while bins 4 and 5 have $x$ values of rather greater than 1, at which stage the particle sizes are within a similar range as the wavelengths. Hence the smallest bins are very insensitive to particle shape, whereas the largest bins display more sensitivity. Within the manuscript, Figure 6 (now Figure 7) has now been adapted to include the variability of the $bpr$ values with aspect ratio for bin 3, and an extra paragraph has been included in the relevant discussion in Section 3 (quoted below). As mentioned in the paragraph introducing Figure 7 in the text, this is the bin with the highest average extinction values.

"*The back-scatter parameter ($bpr$) shown in Figure 7(d) decreases slightly with increasing aspect ratio, especially for larger particles. The phase functions for the smallest bins (not shown) are very symmetric about the sideways scattering angle of $90°$, giving rise to $bpr$ values of 0.49-0.50, values which are insensitive to the aspect ratio. Bin 1 is especially close to the Rayleigh regime for scattering, for these wavelengths. In the case of bin 3, where the scattering parameter $x$ has values between 0.79-1.09 for the three SEVIRI channels, increasing the aspect ratio causes a slight increase in the forward scattering and hence the bpr values decrease slightly. If a particle is particularly forward-scattering, then the outgoing thermal radiation will be comparatively unaffected by it, leading to reduced dust effects simulated at TOA when the bpr values are low.*"

**Page 11 line 27: Wording could be "It is also important"**
6) We agree that adding the word "also" helps to clarify the meaning of this sentence: it has now been added.

**Page 18 line 27: With the mention of needed 3-4 more orders of magnitude of coarse dust, is this mainly due to the smaller mass concentration, smaller area**

[Figure]

**to volume ratio or some other factors? It seems the scattering efficient Q should be high enough for coarse particles.**

7) This is mainly due to the mass concentrations being very low for the coarsest particles in bin 5, however it is clear that the extinction efficiencies are important. The figure on p. C12 shows the extinction efficiencies for the five size bins, for the MSG-3 channels. It is actually not the case that the extinction efficiencies for bin 5 are much greater than for the other bins, in fact at $10.8\,\mu$m the efficiency is greater in bin 4 than in bin 5 for all dust types, for example. Recall that in Equation 7 these efficiencies are divided by the effective radius and multiplied by the mass concentration as simulated by COSMO-MUSCAT. It is also important to recall that the magnitude of the extinction does not give the complete picture, the relationships between the channels are also of vital significance. As a rule-of-thumb, in these plots for a given bin, if the bar at $10.8\,\mu$m is greater than the bars for the other two channels, then the more dust there is in this bin the greater the pink colours that will be produced in the imagery. In bin 5 this spectral contrast is very low for all dust types, in particular compared to the contrasts in bin 3 for the VO73 and SO99 dust types. See also the response to reviewer 2.

**Is it possible that consideration of other channels in the shorter IR and visible wavelengths would further help to constrain aerosol properties with satellite imagery? There are papers that talk about both visible and IR that might be considered as references such as this one: Correlated IR spectroscopy and visible light scattering measurements of mineral dust aerosol. Perhaps this can also be highlighted as future work?**
**https://agupubs.onlinelibrary.wiley.com/doi/full/10.1029/2010JD014389**

8) This is a relevant reference, which we now mention at the beginning of the second paragraph in the introduction: "*While visible and IR channels can provide complementary information about atmospheric dust (e.g. Meland et al., 2010), over bright desert surfaces there are particular challenges for dust imaging and quantification with visible channels.*"

**Referee 2**

**1. The representation of the size distribution of dust and the sensitivity to the coarse fraction is not discussed. In the paper it is shown that the fifth bin above about 8 $\mu$m in diameter as simulated by COSMO-MUSCAT basically do not contribute to the dust load. I wonder how this compares to field observations of the size distribution in particular from FENNEC close to source regions. Moreover, and based on the comparison with field observations, a sensitivity of the results to the coarse size of dust should be added in the paper.**

1) The significance of the size distribution is an important point, which Figure 5 has explored, by means of the integrated extinctions (AOD) subdivided both by refractive indices and by size bin. It is however worth exploring in more detail.

The figure on p. C13 is an extended version of Figure 5, with panel (f) now becoming the new Figure 6 in the updated manuscript. This shows average COSMO-MUSCAT size distributions over BBM, compared with fitted values derived from Fennec measurements. The blue bars are the averages for the bottom 5 km of the atmosphere, a typical daytime boundary layer depth over the Sahara in summer, indicating the likely height-range of well-mixed dust. Meanwhile the red bars are the averages for the bottom 1 km, on the basis that coarse particles especially are not expected to travel high into the atmosphere. It is a valuable comparison to perform, showing that the presence of coarse mode dust in the atmosphere is poorly represented by COSMO-MUSCAT, a general problem for dust transport models.

The coarse mode of dust, as represented by bin 5, is of order $10^6$ times less than that indicated by the measurements. Panel (h) adapts the total in panel (g) to include the total AOD that would be produced if the mass in bin 5 were multiplied by this factor of $10^6$ ('extra coarse'). There is a noticeable slight increase in the AOD values. However if we consider the inter-channel differences in AOD (panels (i) and (j)), which are more
Interactive comment

indicative of the changes in the infrared colour, then this addition of extra coarse dust is not so significant. In both of these panels, the greater the positive value of the channel differences, the stronger pink colours that would arise. Adding extra coarse dust would have the effect of increasing the reduction in the green beam for the VO73 and SO99 dusts, but otherwise the differences are still quite subtle.

We would argue that this sensitivity study is a small distraction to the flow and the overall message of the paper, so we will keep Figure 5 as it is, and add the size distribution plot as a new Figure 6. This figure provides important information about the size distributions going into the radiative transfer simulations, and so we also describe it in Section 3 (quoted below).

"*The average COSMO-MUSCAT dust size distribution over BBM is plotted in Figure 6, displayed in terms of the number density distribution. The blue bars are the averages for the bottom 5 km of the atmosphere, a typical daytime boundary layer depth over the Sahara in summer, indicating the likely height-range of well-mixed dust. Meanwhile the red bars are the averages for the bottom 1 km, exploring the height range that coarser particles are expected to be constrained to. For comparison, lognormal fitted distributions to aircraft measurements made during the Fennec campaign in June 2011 over western areas of the Sahara are overlaid (Ryder et al., 2013). COSMO-MUSCAT bin 3 is broadly in agreement with the fitted measurements, whereas bins 1 and 2 have slightly higher number density values. In contrast, the simulated number density in bin 5 does appear to be too low by several orders of magnitude. That the coarse mode is under-represented is in general an issue in dust modelling, which has been noted before in comparisons with measurements (e.g. Ansmann et al., 2017). Note also, however, that the aircraft measurements display a particularly high degree of variability in the coarse mode compared to the smaller size ranges. Considering Figure 5, if the bin 5 AODs are multiplied by a factor of $10^6$ then these IR AODs would be more equivalent to those simulated in bin 2; however the contrast between the channels is weak (panel (e)), so even such an increase in concentrations would not much perturb the IR colour in the imagery.*"

**2. When comparing the different refractive index datasets in Figure 4, it is evident a strong difference at 10.8 and 12 $\mu$m between Volz, Sokolik, and OPAC data compared to the new laboratory estimates of the RI. This difference may strongly impact the results of this study. I wonder if this is not due to a bias of some of the datasets at some wavelengths. This is for instance discussed for OPAC at 12 $\mu$m due to the large assumed quartz content for this dataset, but I would like a similar discussion also for the strong Volz peak at 10.8 $\mu$m. Is this strong peak reasonable or the dataset is biased/shifted in wavelength compared to the other datasets due to differences in the retrieval procedure or other? I suppose this point should be better addressed given that the sensitivity to the refractive index is the core of the paper.**

2) We agree that further discussion of the potential biases in the refractive indices at various wavelengths would be valuable. Figure 4 has now been enhanced by including two panels which show the refractive index databases at their original spectral resolutions, to indicate the level of variability in their values as a function of wavelength. Superimposed also are the SEVIRI channel filter functions, indicating the significance of their widths. Together with these extra panels we add an extra paragraph of discussion in Section 3 associated with Figure 4, quoted below. With reference to Di Biagio et al. (2017), we discuss some of these differences in the imaginary part of the refractive index. "*Comparing the databases, all of the dust types have peaks in the value of k between approximately 9 and 10 $\mu$m. These peaks lie close to the ozone absorption band, which forms the focus of the SEVIRI 9.7 $\mu$m channel, not used for dust observation purposes due to the ozone absorption at higher altitudes. The VO73 dust has a particularly wide peak, centred at the relatively high wavelength of 10.2 $\mu$m, which causes the convolved value of k to be particularly high in the SEVIRI 10.8 $\mu$m channel. DB17 suggest that the VO73 and OPAC dust types have overestimated k values above 11 $\mu$m, in the quartz absorption band, a feature which may well lead to unreasonably high values in the SEVIRI 12.0 $\mu$m channel. They also note the influence of the clay absorption peak, at ~9.6 $\mu$m, on the wavelength peak*

[Figure]

*of the VO73 dust: this appears to be an important difference between the VO73 and the DB17 dust types, a difference which impacts on the 10.8 μm channel convolved values."*

**3. On the same line, I would appreciate that all the refractive index datasets are included in Fig. 9. I have the impression looking at this figure that the BT is overestimated at 10.8 μm in the Volz dataset and I wonder if this is not linked to a possible bias of the Volz dataset at 10.8 μm compared to all the other estimates of the dust RI used in the paper.**

3) Figure 9 (now Figure 10) with all of the RI databases would produce a very busy plot, but it is possible to create it. This version of the plot is presented on p. C14.

It is apparent from panels (b), (d), and (f) that the DB17 points generally overlap each other in the middle of the plot. The SEVIRI circles, VO73 triangles, SO99 squares, and OPAC diamonds produce the lines which traverse the greatest extent in colour.

We believe that the inclusion of all of the RI databases muddles the plot and makes it less clear, hence we do not propose to include this version of the plot in the paper. However, we are happy to include the DB17-Algeria dataset as an example of the DB17 RIs, and so this will be the new version of Figure 9 (now Figure 10). An extra sentence is included in the manuscript to describe the behaviour of the DB17-Algeria dust on this plot, quoted below:

*"Meanwhile the DB17-Algeria dust has short colour-tracks and hence displays relatively limited colour ranges, approximately following the SO99 pattern over low emissivity surfaces and the VO73 pattern over the higher emissivity surfaces: the limited colour range is an inevitable consequence of the weak extinctions and spectral contrasts apparent in Figure 5."*

It is not clear that it is the VO73 TB values at 10.8 μm which are overestimated, if anything it is possible that they are underestimated compared to the other dust types. These are the solid red bars in the middle of the bar charts (2 < AOD < 3), which are lower than all of the other dust types but still much higher than

the black SEVIRI bars. At the lowest AODs there is a very persistent offset between the high SEVIRI and the low simulation TBs, probably a consequence of low biased skin temperatures in the simulations (see original manuscript p.14, lines 20-24).

**The right panel of figure 9 is not very clear to me; please, better explain its content and message.**
4) Figure 9 (now Figure 10) is indeed a very busy plot, which is discussed at length in Section 5.2. We have tried to define the concept of the figure completely in the caption, however in the third paragraph of Section 5.2 we now include the sentence: "*Each point in these panels indicates the average colour that is produced from a subset of brightness temperature data, for the specified dust type and within the specified ranges of AOD and surface emissivity.*" We hope that this makes the core purpose of the figure clearer, since we suspect that this is the missing link in the explanation. For extra clarity, we also now mark the AOD ranges corresponding to each colour on panel (d).

**I agree with the other reviewer in asking to add the word "infrared" in the title.**
5) As with point 1 by reviewer 1, we very much agree that the word 'infrared' should be in the title. It is now included.

[Figure]

**Fig. 1.** Phase functions as a function of particle size and wavelength (see p. C3).

[Figure]

[Figure]

**Fig. 2.** Extinction efficiency as a function of particle size and wavelength (see p. C5).

[Figure]

**Fig. 3.** IR AODs including extra coarse dust (see p. C6 for details).

[Figure]

**Fig. 4.** Update to Figure 9 in the ACPD version, including more dust types (see p. C9 for details).

---

## Author Response (AR2)

**Responses to editor's comments on "The influence of dust optical properties on the colour of simulated MSG-SEVIRI Desert Dust infrared imagery"**

We thank very much the editor for her very helpful comments, which have added informative extra insight into the paper. Below are our responses and our changes, with the editor's comments included in bold.

**Abstract – It would be appropriate to mention the choice and impacts of horizontal alignment in the non-sphericity calculations, since this is a fairly novel and important application on the scale they authors employ in this study.**
A couple of sentences on this have been added to the abstract:
"The consequences for the infrared extinctions of both the particle shape and the particle orientation are explored: this analysis shows that as the particle asphericity increases, the extinctions increase if the particles are aligned horizontally, and decrease if they are aligned vertically. Randomly oriented spheroidal particles have very similar infrared extinction properties as spherical particles, whereas the horizontally and vertically aligned particles can be considered to be the upper and lower bounds on the extinction values. Inputting these values into COSMO-MUSCAT-RTTOV, it is found that spherical particles..."

**P3 l1 – 'output' – please specify what specifically e.g. dust mass, size, location, altitude etc.**
This has now been clarified using the words 'four-dimensional (space and time) and size-resolved dust mass concentrations' before the word 'output'.

**Section 2.2, l18-23 – are the simulations nudged/what boundary conditions are used? ERA-interim? Are dust-radiative feedbacks on atmospheric circulation included?**
Dust-radiative feedbacks are included, and the model is driven by boundary conditions provided by output from the global model GME of Deutscher Wetterdienst. In the text we now add the following sentences:
"The COSMO model is driven by initial and boundary data from analysis fields of the former global model GME of the German weather service (Deutscher Wetterdienst, DWD). The simulations include an online feedback of modelled dust on the solar and thermal radiation fluxes in order to account for the dust radiative impact on atmospheric dynamics. In the radiation computations, the dust optical properties are considered as described above."

**P10 l 16-28 – Concentrations in bins 1 and 2 are also fairly high in the model compared to observations, which is another common challenge for models (e.g. Kok et al., 2017, Nature Geosciences). It would be interesting if the authors could also speculate/quantify the impacts on pinkness of scaling down the concentrations in bin 1 and 2, in the same way they illustrate this for scaling up bin 5.**
The Kok et al. (2017) reference has now been included to highlight the over-representation of the fine mode. Some extra speculation is added to the end of this paragraph:
"If the smaller dust particles in bins 1 and 2 had their concentrations reduced by a factor of 10, then it is to be expected for the VO73 and SO99 dust types that the red beam would be slightly reduced and the green beam very slightly increased."
The comparable plot for the scaling-down of bins 1 and 2 ('less fine') is included at the end of these author comments.

**P10/11 l29-3, aspect ratios. Please can the authors clarify which orientation their AR is specified in for oblate spheroids. They should also note that few studies have retrieved the 3-D AR such as done by Chou et al., 2008, who found a height-to-maximum-axis ratio of 0.4, besides the AR value of 1.7. Is a 3-D AR considered here, or are two of the main ellipsoid axes the same length?**
The ARs considered here are only 2-D, so two of the main ellipsoid axes will be the same length. The oblate spheroids are oriented horizontally. In the T-matrix code used, the 'axis ratio' is the ratio of the 'horizontal-to-vertical' axes, which is identical to the aspect ratio as defined in Section 3 (ratio of the semi-major to the semi-minor axes) if the spheroid is horizontally oriented. If the spheroid is vertically oriented and rotating around the vertical axis (prolate), then the axis ratio would be equal to the inverse of the aspect ratio.

**P11 l15-28 – Fig 7 does not appear to show results for spheres or vertically aligned particles. If this is the case, please state 'not shown' in the text to clarify.**

We have now added the vertically aligned VO73 particles to Figure 7(c), denoted by the small filled triangles, whose extinction values descend with increasing aspect ratio. The corresponding paragraph has been amended slightly to take more account of the vertically aligned particles. Spheres are represented by the values at AR = 1.0.

**P11 l15-28 – What is the reasoning behind only taking forward the case of horizontally aligned particles, given that vertically aligned also provide a lower bound, and both appear equally likely?**

As above, we now include slightly more discussion on the vertically aligned case, enhancing Figure 7(c) with this information. Figures 10 and 11 provide some insight into this: Figure 10 indicates that the spherical case does not appear to produce the range of colour values that is seen in the SEVIRI observations/retrievals, and it is also clear in Figure S4 (in the Supplement, see below) that the relatively weak extinctions of the DB17 dust types give rise to weaker colour responses. In Figure 11, increasing the asphericity of horizontally aligned dust brings the red and green beams closer to those provided by SEVIRI. It is clear that vertically aligned dust would have the opposite effect, and would have a weaker colour response.

**P12 l4-10 – It is possible to cite ACPD discussion reviews – please do so here since the additional plots in the author response are valuable to the interpretation of the article (or put the plots in a supplement).**

A good suggestion, the four plots in the author response from 31st May 2018 have now been included in a Supplement document. This particular plot is Figure S3.

**P12 l30 – please add Ouarzazate and Tamanrasset locations to the map on Figure 1**

These are also relevant locations, and have now been included.

**P14 l31 paragraph – What is the reasoning behind returning to a spherical assumption here in Fig 10?**

As indicated in Figure 7, the spherical assumption lies approximately in the middle of the range of output extinction properties, between the horizontally and vertically aligned cases. Given the lack of certainty on either the particle shape or orientation, and whether there should be any preferred values of these at all, the spherical assumption seems to be the most justified at this stage. Figure 10 is more focussed on the differences between dust types, so for clarity only one particle shape is considered, for which the spherical case can be regarded as the default.

It is possible to re-create Figure 10 with different particle shapes, but this would extend the paper for little extra benefit; as an example, however, Figure 10 is reinterpreted in Figure S5, with horizontally oriented dust with an AR value of 1.7. Due to the increased extinctions, the VO73 and SO99 colour tracks are slightly longer in Figure S5 than they are in Figure 10.

**P14 l35 – 'Saharan domain' – please give latitude/longitudes here. Is the Sahel included?**

This has now been clarified in the text. The domain covers 12-36N and 18W - 35E, which also includes most of the Sahel.

**P15 l18 – 'left set' – i.e. hatched bars?**

The word 'left' has now been replaced with the word 'hatched'.

**P16 l31-34 – Similar to my comment above, the additional plots of the BT differences for the extra coarse case are useful to the interpretation of the paper. Please either cite the author response, or add the plots to a supplement.**

This has been added to the Supplement, Figure S2.

**p 20 l16 – this may be particularly pertinent since dust and moisture are often correlated at dust emission during Saharan summer (e.g. Marsham et al., 2013, 2016).**

A relevant point, and citation, which has now been added. Hence the penultimate sentence, leading into the final sentence:
"Analysis by Marsham et al. (2016) of observational data from the central Sahara has suggested that variations in water vapour dominate the variability in net TOA radiation (water vapour also exhibits some correlation with dust AOD), indicating the significance of water vapour for satellite observations over regions with high dust activity. Given these influences of water

vapour and atmospheric temperature on the TOA signal, it is clear that the characteristics..."

**Data availability – It would be useful to provide the SO99 refractive index dataset in some way, since they do not appear available via the Helmert 2007 publication.**

5    We now state in the first sentence of the Data availability section that these are available on request from the authors, along with the COSMO-MUSCAT and RTTOV output.

**Table 3 – please expand 'SEV' and 'AER' for clarity in the caption.**

This has now been clarified in the caption.

**Fig 11 – please add to caption what the left/right columns represent.**

'left column' and 'right column' have been added in parentheses after the words 'brightness temperatures' and 'RGB colours', respectively.

15    We have very recently noticed an error in the treatment of the emissivity subsets in Section 5.2, affecting Figures 10 and 11. In the analysis code that produced these plots, only June emissivity values had been included, and so July TB and colour values had been referenced to June emissivity values. Functionally this means that slightly more points in these comparisons belong in the higher emissivity ranges. Re-running this program with the correct co-located June/July and MSG-2/MSG-3 emissivities gives updated Figures 10 and 11, and updated numbers in Section 5.2. The differences are negligible but non-zero, and several

20   numbers have been changed very slightly: these changes do not affect the overall discussion.

[revised manuscript text omitted]